# Penalizing Infeasible Actions and Reward Scaling in Reinforcement Learning with Offline Data

## Abstract

Reinforcement learning with offline data often suffers from Q-value extrapolation errors due to limited data, which poses significant challenges and limits overall performance. Existing methods such as layer normalization and reward relabeling have shown promise in addressing these errors and achieving empirical improvements. In this paper, we extend these approaches by introducing reward scaling with layer normalization (RS-LN) to further mitigate extrapolation errors and enhance performance. Furthermore, based on the insight that Q-values should be lower for infeasible action spaces—where neural networks might otherwise extrapolate into undesirable regions—than for feasible action spaces, we propose a penalization mechanism for infeasible actions (PA). By combining RS-LN and PA, we develop a new algorithm called PARS. We evaluate PARS on a range of tasks, demonstrating superior performance compared to state-of-the-art algorithms in both offline training and online fine-tuning across the D4RL benchmark, with notable success in the challenging AntMaze Ultra task.

## 1 Introduction

Reinforcement learning (RL) enables agents to develop optimal decision-making strategies through real-time interactions. However, these interactions with real-world environments can expose the agent to considerable risks. To mitigate these risks, Offline RL, which derives optimal policies from pre-collected data, has emerged as a critical area of research (Fujimoto & Gu, 2021; Tarasov et al., 2024). Additionally, agents trained with offline RL can be deployed in real-world environments to further acquire knowledge, leading to the development of offline-to-online RL approaches (Lee et al., 2022; Nakamoto et al., 2024; LEI et al., 2024). However, due to the limited coverage of offline data, these methods often suffer from extrapolation error, where the Q-values of out-of-distribution (OOD) actions are overestimated, limiting the overall performance (Kumar et al., 2020; Kostrikov et al., 2022; Lyu et al., 2022; Mao et al., 2024).

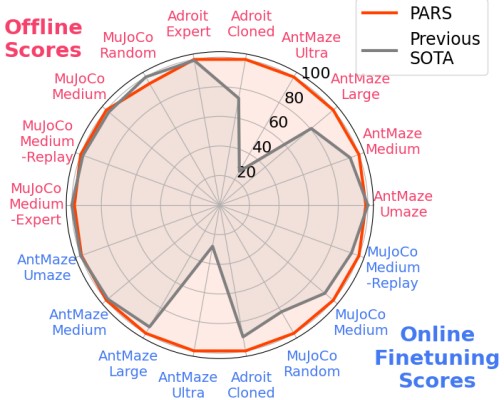

Figure 1: Overview of the comparison between PARS and prior SOTA with scores normalized to each task's maximum performance.

To improve the performance of RL with offline data, several methods have been proposed. Recent work by Ball et al. (2023) has demonstrated that layer normalization (LN, Ba et al. (2016)) can reduce catastrophic overestimation by constraining Q-values in OOD actions, promoting more stable Q-value learning and enhancing overall performance. Additionally, from a different perspective, reward relabeling has been applied in various offline RL algorithms. For example, in sparse reward environments like AntMaze, Kostrikov et al. (2022) modifies the rewards by subtracting 1, while Tarasov et al. (2024) multiplies all rewards by 100.

In this work, we examined the impact of these two techniques on RL with offline data and demonstrated that combining layer normalization with reward scaling (applying a constant factor to the overall reward) more effectively alleviates extrapolation error. However, we observed that the Q-function still tends to be overestimated beyond regions where data coverage ends. To further address this, we penalize the Q-values of infeasible action regions distant from the agent's feasible action regions, which encourages a gradual reduction in Q-values beyond the data-covered regions.

Building on these insights, we introduce PARS (**P**enalizing infeasible **A**ctions and **R**eward **S**caling), a novel algorithm designed to mitigate Q-function extrapolation error. PARS consists of two components: **(1) reward scaling with layer normalization (RS-LN)** and **(2) penalizing infeasible actions (PA)**. PARS is based on a minimal approach for offline RL, TD3+BC (Fujimoto & Gu, 2021), with RS-LN and PA being implementable through just a few lines of code. This simplicity allows PARS to be easily integrated into existing off-policy algorithms with minor modifications, making it highly practical for real-world use.

We evaluate PARS on various RL tasks during both the offline training and online fine-tuning phases. Figure 1 provides an overview of the benchmark comparison, showing that the performance of PARS is either close to or significantly better than the previous SOTA in all considered tasks. In particular, PARS stands out as the only method to successfully learn during offline-to-online training in the challenging Antmaze Ultra task, demonstrating strong performance.

## 2 PRELIMINARIES

The RL problem is formulated as a Markov Decision Process (MDP, Puterman, 1990) $\mathcal{M} = \langle \rho_0, \mathcal{S}, \mathcal{A}, P, \mathcal{R}, \gamma \rangle$, where $\rho_0$ is the initial state distribution, $\mathcal{S}$ is the state space, $\mathcal{A}$ is the action space, $P(s_{t+1}|s_t, a_t)$ is the transition probability, $\mathcal{R}(s_t, a_t)$ is the reward function, and $\gamma \in (0, 1)$ is the discount factor, with $s_t \in \mathcal{S}$ and $a_t \in \mathcal{A}$ denoting the state and action at timestep $t$, respectively. In this study, we focus on a continuous action space, typically confined to a compact subset of $\mathbb{R}^n$. We denote the action space $\mathcal{A}$, as defined in the MDP, as the **feasible action region** $\mathcal{A}_\mathcal{F}$, and the **infeasible action region** as $\mathcal{A}_\mathcal{I} = \mathbb{R}^n \setminus \mathcal{A}_\mathcal{F}$, which consists of actions the agent cannot perform in any state. We extend the action space, typically limited to feasible actions, by defining the infeasible action space $\mathcal{A}_\mathcal{I}$ to account for potential extrapolation into infeasible regions by neural networks.

Given this formulation, offline RL allows for learning a policy without real-time interaction with the online environment and enables learning from a pre-collected dataset $\mathcal{D}$, composed of trajectories of various quality, $\tau = \{(s_t, a_t, r_t, s_{t+1})\}_{t=0}^{T-1}$, with $r_t = \mathcal{R}(s_t, a_t)$ denoting the reward at timestep $t$ and $T$ representing the episode length, without knowing the behavior policy used to collect $\mathcal{D}$. The goal of offline RL is to utilize the transitions in $\mathcal{D}$ to develop an optimal policy $\pi$ that maximizes the expected discounted return $\sum_{t=0}^{T-1} \gamma^t r_t$.

A policy $\pi$ trained with $\mathcal{D}$ can be finetuned to suit online situations by interacting with the online environment. As new transitions are collected in a replay buffer $\mathcal{B}$ through online interactions, the policy can be further improved by sampling transitions from both $\mathcal{D}$ and $\mathcal{B}$.

**Layer normalization.** Layer normalization (LN; Ba et al., 2016) stabilizes neural network training by normalizing hidden outputs $\{x_i\}$ in each $i$-th layer. LN re-centers and re-scales them using the layer's mean $\mu$ and standard deviation $\sigma$ with the transformation $\hat{x}_i = \frac{x_i - \mu}{\sigma}\eta + \beta$, where $\eta$ and $\beta$ are learnable parameters. In recent RL research, LN has been shown to improve training stability and thus lead to increased final performance (Ball et al., 2023; Nauman et al., 2024). Notably, Ball et al., 2023 demonstrates that applying LN can constrain Q-values for out-of-distribution actions by the weight layer norm, thereby mitigating the effects of erroneous Q-function extrapolation.

## 3 CRITIC REGULARIZATION FOR OOD ACTIONS

When RL is performed solely using a dataset without any online interaction, the Q-function may become unstable and diverge due to overestimation in the OOD action region (Kumar et al., 2020; An et al., 2021; Yue et al., 2024), as shown in Figure 2 (a). To address this issue, various critic regularization methods have been proposed including the ensemble approach, penalizing OOD action within the feasible region. Figure 2 captures the essence of the nature of the realistic offline dataset.

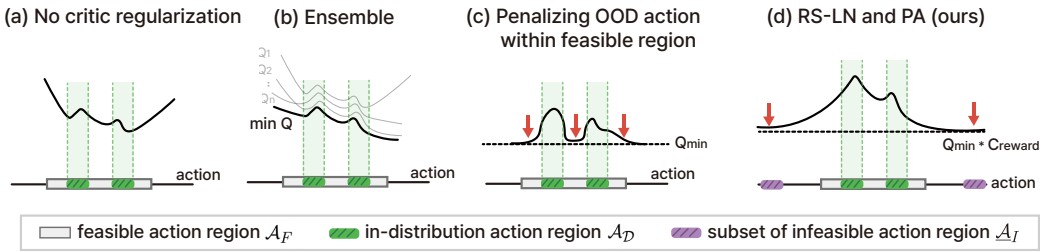

Figure 2: Conceptual comparison of various critic regularization methods (a-c), including our approach with RS-LN and PA (d).

There exists a feasible action region $\mathcal{A}_F$, and there exists an in-distribution (ID) region $\mathcal{A}_D$ ($\subset \mathcal{A}_F$), which may consist of multiple discontiguous subregions.

First, An et al. (2021) showed that increasing the number of critics in an ensemble can provide effective critic regularization, as depicted in Figure 2 (b). Additionally, other works (Kumar et al., 2020; Lyu et al., 2022; Mao et al., 2024) introduce a penalty to reduce Q-values for OOD actions deviating from the behavior policy $\mu$ within the feasible action space $\mathcal{A}_F$, as shown in Figure 2 (c). For instance, MCQ (Lyu et al., 2022) and SVR (Mao et al., 2024) use behavior models trained with either a VAE or Gaussian distribution to distinguish OOD actions and penalize them.

However, both approaches have limitations. Achieving sufficient regularization using only a critic ensemble requires a large number of critics, significantly increasing training complexity. Furthermore, it does not perform well on sparse reward tasks, such as AntMaze (Tarasov et al., 2022; 2024). Meanwhile, methods that rely on behavior models are limited by their accuracy. Misclassifying ID actions as OOD can lead to inappropriate penalties and underestimated Q-values. Additionally, considering online fine-tuning, adapting the behavior model in an evolving online setting presents further challenges (Nair et al., 2020). Moreover, exploring potential regions not present in the dataset and adapting accordingly becomes difficult if the OOD action region within $A_F$ is fitted to a specific penalty value or is overly conservative. Appendix B provides a more comprehensive comparison.

Therefore, we propose an alternative viewpoint on critic regularization. Unlike previous methods, we leverage layer normalization and reward scaling to enhance the network's expressivity, which naturally mitigates OOD overestimation. Additionally, rather than penalizing OOD actions within $A_F$, our method imposes penalties on OOD actions outside $A_F$ to further prevent overestimation in extrapolation regions, as shown in Figure 2 (d). Our goal through this is to smoothly reduce the values of OOD actions outside the data coverage while seamlessly interpolating the values of OOD actions within the data coverage. This allows for flexible adaptation to distribution shifts during online fine-tuning (See Appendix B). Moreover, our method can be combined with a small number of ensembles to further stabilize critic regularization.

## 4 PENALIZING INFEASIBLE ACTIONS AND REWARD SCALING

### 4.1 DIDACTIC EXAMPLE

We first examine the effects of reward scaling in conjunction with LN and the penalization of infeasible action regions (PA) when learning a Q-function with function approximation. For this, we considered a regression problem by adopting and modifying the approach used in Ball et al. (2023). The true Q-function is defined as

$$y = f(x_1, x_2) = c_{\text{reward}} \cdot \left( \sqrt{x_1^2 + x_2^2} \cdot \cos\left(\pi/8\right) + x_1 \cdot \sin\left(\pi/8\right) \right),$$

where $c_{\text{reward}}$ is the reward scaling factor and the feasible input region is defined as $(x_1, x_2) \in [-1, 1]^2$. This function is a slanted, inverted cone-shaped function with 2D inputs, $x = (x_1, x_2)$, as shown in Figure 3. We then generated a dataset $\{(x_1, x_2, y)\}$ as $y = f(x_1, x_2)$ only on the ID region $x_1^2 + x_2^2 \le 0.5^2$, which covers only a subset of the feasible input region. The data were then fitted using a 3-layer MLP with a hidden dimension of 256 and ReLU activation (Agarap, 2018).

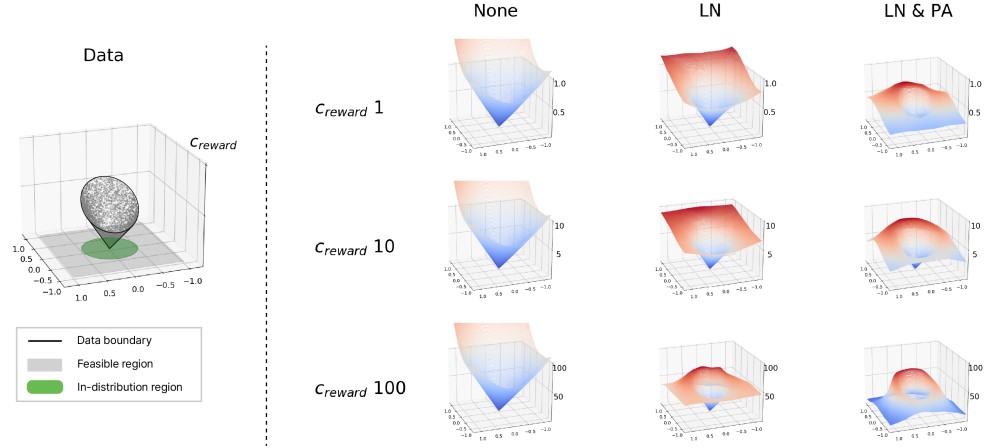

Figure 3: Results of training on a toy dataset using MLP networks with vanilla regression ('None' column), with LN ('LN' column) and LN with PA ('LN & PA' column), varying the $c_{\text{reward}}$.

For this regression fitting, we considered three cases: 1) vanilla regression, 2) regression with LN, and 3) regression with LN and PA. PA is applied so that penalties for infeasible inputs are imposed by setting the label $y$ to 0 for inputs where $x_1$ or $x_2 \in (-200, -100) \cup (100, 200)$, which are far from the feasible input region. By imposing penalties on regions far from the feasible region, the impact on the prediction of Q-values within the feasible region can be minimized. The experiments were conducted by varying the reward scaling factor $c_{\text{reward}}$ among 1, 10, and 100.

As shown in the 'None' column of Figure 3, when the dataset covers only a portion of the feasible region rather than the full region, Q-value regression yields overestimation, particularly from the points where dataset coverage ends. Integrating LN into the Q network helps mitigate this abrupt overestimation of Q-values as previously noted by Ball et al. (2023), with its impact becoming more pronounced as the reward scale increases ('LN' column). Additionally, combining PA with LN enables the network to decrease gradually in regions where data coverage ends as the input norm grows ('LN & PA' column). Note that reward scaling is effective only when used in conjunction with LN; when used independently, it can lead to the divergence of Q-values. For more discussion on the didactic example, please refer to Appendix C.

### 4.2 PARS Algorithm

Based on insight from Section 4.1, we present a novel algorithm that prevents Q-value extrapolation error to ensure stable Q-learning across both offline and online fine-tuning phases: **P**enalizing in-feasible **a**ctions and **r**eward **s**caling (PARS). PARS is based on the minimalist offline RL algorithm, TD3+BC, and is built upon two key components: **(1) reward scaling combined with LN (RS-LN)** and **(2) penalizing infeasible actions (PA)**.

#### 4.2.1 PARS Component 1: Reward Scaling Combined with LN (RS-LN)

For our first component, RS-LN, we scale all rewards in the offline dataset $\mathcal{D}$ by multiplying each reward $r \in \mathcal{D}$ by a constant factor $c_{\text{reward}}$. During online fine-tuning, the rewards stored in the replay buffer $\mathcal{B}$ are similarly scaled by the same constant factor $c_{\text{reward}}$, ensuring consistency across both offline and online phases. In the previous didactic example (Figure 3), we showed the effectiveness of RS-LN in mitigating OOD values, and we aim to further investigate potential underlying causes.

Referring to Figure 3, RS-LN appears to foster the learning of a more intricate decision boundary. This observation leads to the hypothesis that mitigating OOD values is linked to the network's capacity to learn more complex functions. This capacity, often referred to as neural network expressivity, has been extensively studied in various RL contexts (Kumar et al., 2021; Sokar et al., 2023; Obando Ceron et al., 2024), and several metrics to approximate expressivity have been proposed. Kumar et al. (2021) introduced the effective rank, $\text{Srank}_\delta(\phi)$, which provides insights into how efficiently the model utilizes its capacity to represent complex data. A low $\text{Srank}_\delta(\phi)$ suggests the possibility of under-parameterization. Additionally, Sokar et al. (2023) introduced the concept of

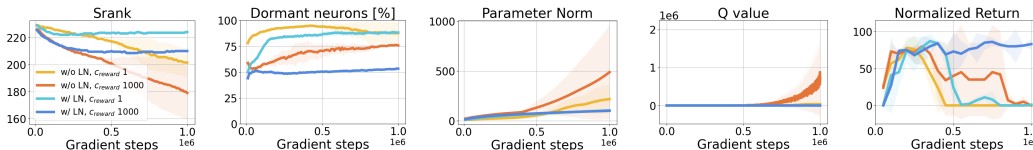

Figure 4: Plots illustrating how each metric changes during training in AntMaze-medium-diverse-v2, comparing the cases with (w/) and without (w/o) LN for $c_{\text{reward}}$ values of 1 and 1000.

dormant neurons (neurons with zero activations) in RL, showing that an increase in dormant neurons correlates with the network's under-utilization and degraded performance.

Figure 4 shows how each metric evolves during training as $c_{\text{reward}}$ increases, both with and without LN. Without LN, the parameter norm grows rapidly as $c_{\text{reward}}$ increases, and $\text{Srank}_\delta(\phi)$ decreases sharply. Conversely, with LN, the parameter norm remains controlled, and although $\text{Srank}_\delta(\phi)$ decreases initially, it converges to a stable value. This regulated reduction in $\text{Srank}_\delta(\phi)$ prevents overfitting by minimizing the learning of irrelevant noise, allowing the model to focus on key patterns, thus enhancing performance (Obando Ceron et al., 2024). Dormant neurons are also more effectively utilized as $c_{\text{reward}}$ increases with LN. Therefore, increasing $c_{\text{reward}}$ with LN helps the network focus on important information, boosts network expressivity, mitigates OOD value spikes, and ultimately leads to performance improvements.

### 4.2.2 PARS COMPONENT 2: PENALIZING INFEASIBLE ACTIONS (PA)

The goal of penalizing infeasible actions is to prevent $Q$ values from diverging from the point where data coverage ends. To accomplish this, in addition to the standard TD loss, we consider the PA loss to constrain the value in the infeasible action region to $Q_{\min}$. Since we want the Q-function in the feasible region to be sufficiently well estimated with the dataset and not heavily affected by the constraint in the infeasible action region, we allow some guard interval between the feasible action region and the constraint-imposed infeasible action region.

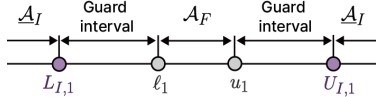

Figure 5: $\mathcal{A}_F$ and $\underline{\mathcal{A}}_I$ for $n = 1$

Therefore, we consider the following subset of the infeasible action region:

$$\underline{\mathcal{A}}_I = \bigcup_{i=1}^{n} \{(-\infty, L_{I,i}] \cup [U_{I,i}, \infty)\}, \tag{1}$$

where $n$ is the dimensions of action. Note that the feasible action region is defined as $\bigcap_{i=1}^{n} \{(\ell_i, u_i)\}$, where $\ell_i$ and $u_i$ denote the lower and upper bounds of the feasible space in the $i$-th dimension. Thus, $L_{I,i} < \ell_i$ and $u_i < U_{I,i}$ to allow the guard or transition interval. Then, the PA loss is determined as

$$\mathcal{L}_{\text{PA}} = \min_\phi \mathbb{E}_{s \sim \mathcal{D}, a \in \underline{\mathcal{A}}_I} \left[ (Q_\phi(s, a) - Q_{\min})^2 \right], \tag{2}$$

where $Q_{\min}$ can be calculated as $c_{\text{reward}} \cdot r_{\min}/(1-\gamma)$. If the minimum reward of the task is unknown, it can be estimated from the dataset's minimum reward, as suggested in Mao et al. (2024). By combining two losses with a weighting factor $\alpha$, our modified TD loss is given as follows:

$$\mathcal{L}_{\text{Total}} = \min_\phi \mathbb{E}_{s,a,s' \sim \mathcal{D}} \left[ \left( Q_\phi(s, a) - \left( c_{\text{reward}} \cdot r(s, a) + \gamma \mathbb{E}_{a' \sim \pi_\theta(\cdot|s')} Q_\phi(s', a') \right) \right)^2 \right]$$
$$+ \alpha \cdot \left( \min_\phi \mathbb{E}_{s \sim \mathcal{D}, a \in \underline{\mathcal{A}}_I} \left[ (Q_\phi(s, a) - Q_{\min})^2 \right] \right) \tag{3}$$

### 4.2.3 IMPLEMENTATION

**Infeasible action sampling.** The expectation over $a \in \underline{\mathcal{A}}_I$ in eq. (2) is practically realized by sample expectation or sample mean. To sample actions from the support in eq. (1), we consider the following uniform distribution for infeasible action sampling:

$$a_i \sim \begin{cases} \text{Uniform}(L_{I,i} - \Delta_L, \ L_{I,i}) & \text{for } a_i \leq L_{I,i} \text{ with probability } 0.5, \\ \text{Uniform}(U_{I,i}, \ U_{I,i} + \Delta_U) & \text{for } a_i \geq U_{I,i} \text{ with probability } 0.5. \end{cases} \tag{4}$$

As we shall see in Section 6.4, the performance does not heavily depend on the values of $L_{I,i}$, $U_{I,i}$ when these values are set as 100 to 1000 times the boundary of the feasible region.

**Critic ensemble.** PARS can be integrated into an ensemble approach with a limited number of critics (4 for AntMaze, 10 for MuJoCo and Adroit). The impact of the number of critics is discussed in Appendix H. For policy evaluation, we use the same approach as in Ball et al. (2023) for both offline and online fine-tuning. However, for policy improvement, we employ slightly different objectives. During offline training, we use the minimum critic value. For online fine-tuning, we average a subset of critics with a sample size of $k_a$ to avoid restricting the online exploration. The objective function for policy improvement is as follows, where $\beta$ is the weight factor for the behavior cloning regularization term in TD3+BC, and $\lambda = \frac{1}{\mathbb{E}(|\min Q_{\phi_j}(s,a)|)}$:

$$\text{Offline training:} \quad \max_{\theta} \mathbb{E}_{s,a\sim\mathcal{D}} \left[ \lambda \left( \min_{j=1,...,N} Q_{\phi_j}(s, \pi_\theta(s)) \right) - \beta \cdot (\pi_\theta(s) - a)^2 \right], \quad (5)$$

$$\text{Online fine-tuning:} \quad \max_{\theta} \mathbb{E}_{s,a\sim\mathcal{D}} \left[ \lambda \left( \mathbb{E}_{j\in\mathcal{Z}_{k_a}} Q_{\phi_j}(s, \pi_\theta(s)) \right) - \beta \cdot (\pi_\theta(s) - a)^2 \right] \quad (6)$$

## 5 RELATED WORKS

**Offline RL methods.** In addition to the offline RL methods discussed in Section 3, which focus on critic regularization, there is also ongoing research into actor regularization aimed at preventing the actor from straying too far from the behavior policy. For example, TD3+BC (Fujimoto & Gu, 2021), an offline variant of TD3 (Fujimoto et al., 2018), adds a behavior cloning term to the policy objective to keep $\pi$ close to the behavior policy. Additionally, SPOT (Wu et al., 2022) introduces a density-based regularization term using a VAE to estimate the behavior policy's density, explicitly modeling the support set. Furthermore, in an approach utilizing both actor and critic regularization, ReBRAC (Tarasov et al., 2024) recently enhanced TD3+BC by incorporating previously unvalidated design elements, including decoupling the actor and critic penalties. ReBRAC demonstrates that significant performance gains can be achieved by integrating appropriate design choices into existing methods.

**Offline-to-online RL methods.** The policy $\pi$, trained using the offline dataset, can be further fine-tuned through additional online interaction. However, this new interaction often leads to distributional shifts between the offline and online data, resulting in performance degradation (Nair et al., 2020; Lee et al., 2022; Uchendu et al., 2023). To address this issue, various studies have explored effective fine-tuning methods for offline algorithms (Lee et al., 2022; Nakamoto et al., 2024; Zhang et al., 2023; Beeson & Montana, 2022). For example, Off2On (Lee et al., 2022) attempts to mitigate distribution shift issues by using an ensemble and a balanced replay strategy based on CQL (Kumar et al., 2020). However, traditional offline algorithms are designed with limited datasets in mind, leading to a conservative learning approach that can limit performance in online fine-tuning. To overcome this, Uni-O4 (LEI et al., 2024) proposes removing conservatism during the offline phase to facilitate a smoother transition to the online phase. On the other hand, RLPD (Ball et al., 2023) takes a different approach by bypassing the explicit offline phase. Instead, it leverages offline data during online learning, utilizing LN and increasing the update-to-data (UTD) ratio for efficient sample utilization. Recently, Zhao et al. (2024) proposed ENOTO, effectively utilizes ensembles for efficient offline-to-online RL, showcasing its effectiveness across various ensemble-based approaches. Compared to existing methods, we do not propose new specialized techniques for online fine-tuning. RS-LA and PA naturally integrate into the process with the critic ensemble, enabling smooth transitions and superior performance for PARS.

## 6 EXPERIMENTS

### 6.1 EXPERIMENT SETUP

**Benchmark.** We use three domains (AntMaze, Adroit, and MuJoCo) with a total of 28 datasets from the D4RL benchmark (Fu et al., 2020) for our experiments. The view of example datasets can be found in Figure 6. In the performance comparison table, we used the following abbreviations: u for umaze, m for medium, l for large, p for play, d for diverse, r for random, m-r for medium replay, and e for expert. For a detailed explanation of the benchmark please refer to Appendix G.1.

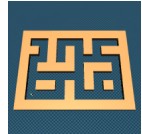 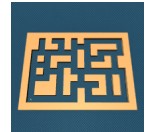 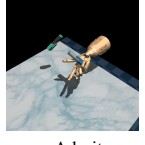 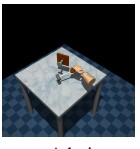 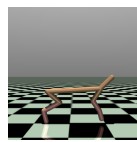 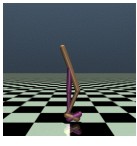

(a) AntMaze Umaze    (b) AntMaze Large    (c) AntMaze Ultra    (d) Adroit Pen    (e) Adroit Hammer    (f) MuJoCo HalfCheetah    (g) MuJoCo Walker2d

Figure 6: Examples of datasets we used for experiments.

**Hyperparameters.** We set $L_I$ and $U_I$ uniformly across all action dimensions as $L_{I,i} = L_I$ and $U_{I,i} = U_I$ for each dimension $i$. We use $\Delta_L = |L_I|$ and $\Delta_U = |U_I|$, where $L_I$ and $U_I$ are scaled by either 100 or 1000 times the boundary of the feasible region. Given that in the tasks considered, $L_I < 0$ and $U_I > 0$, we sample infeasible actions from the intervals $[2L_I, L_I]$ and $[U_I, 2U_I]$, respectively. Furthermore, we enable the tuning of around 10 sets of hyperparameters, including $\alpha$ and TD3+BC's $\beta$, following prior works (Wu et al., 2022; Nikulin et al., 2023; Tarasov et al., 2024). For detailed hyperparameters and implementation, please refer to Appendix G.3.

Table 1: Offline PARS evaluation on the AntMaze, Adroit, and MuJoCo domains. We report the final normalized score averaged over five random seeds, with ± representing the standard deviation.

| AntMaze | TD3+BC | IQL | CQL | SAC-N | EDAC | MCQ | MSG | SPOT | SAC-RND | ReBRAC | PARS |
|---|---|---|---|---|---|---|---|---|---|---|---|
| antmaze-u | 78.6 | 87.5 | 74.0 | 0.0 | 0.0 | 27.5 | **97.9** | 93.5 | **97.0** | **97.8** | 93.8±2.1 |
| antmaze-u-d | 71.4 | 62.2 | 84.0 | 0.0 | 0.0 | 0.0 | 79.3 | 40.7 | 66.0 | 88.3 | **89.9**±7.5 |
| antmaze-m-p | 10.6 | 71.2 | 61.2 | 0.0 | 0.0 | 0.0 | 85.9 | 74.7 | 38.5 | 84.0 | **91.2**±3.9 |
| antmaze-m-d | 3.0 | 70.0 | 53.7 | 0.0 | 0.0 | 0.0 | 84.6 | 79.1 | 74.7 | 76.3 | **92.0**±2.2 |
| antmaze-l-p | 0.2 | 39.6 | 15.8 | 0.0 | 0.0 | 0.0 | 64.3 | 35.3 | 43.9 | 60.4 | **84.8**±5.9 |
| antmaze-l-d | 0.0 | 47.5 | 14.9 | 0.0 | 0.0 | 0.0 | 71.2 | 36.3 | 45.7 | 54.4 | **83.2**±5.6 |
| antmaze-ultra-p | 0.0 | 13.3 | 16.1 | 0.0 | 0.0 | 0.0 | 0.6 | 4.4 | 20.6 | 22.4 | **66.4**±4.4 |
| antmaze-ultra-d | 0.0 | 14.2 | 6.5 | 0.0 | 0.0 | 0.0 | 1.0 | 12.0 | 10.5 | 0.8 | **51.4**±11.6 |
| average | 20.5 | 50.7 | 40.8 | 0.0 | 0.0 | 3.4 | 60.6 | 47.0 | 49.6 | 60.6 | **81.6** |

| Adroit | TD3+BC | IQL | CQL | SAC-N | EDAC | MCQ | SVR | SPOT | SAC-RND | ReBRAC | PARS |
|---|---|---|---|---|---|---|---|---|---|---|---|
| pen-cloned | 61.5 | 77.2 | 39.2 | 64.1 | 68.2 | 35.3 | 65.6 | 15.2 | 2.5 | 91.8 | **107.5**±15.8 |
| pen-expert | 146.0 | 133.6 | 107.0 | 87.1 | -1.5 | 121.2 | 119.9 | 117.3 | 45.4 | **154.1** | 152.7±1.0 |
| door-cloned | 0.1 | 0.8 | 0.4 | -0.3 | **9.6** | 0.2 | 1.1 | 0.0 | 0.2 | 1.1 | 4.3±6.1 |
| door-expert | 84.6 | 105.3 | 101.5 | -0.3 | **106.3** | 73.0 | 83.3 | 0.2 | 73.6 | 104.6 | 106.0±0.2 |
| hammer-cloned | 0.8 | 1.1 | 2.1 | 0.2 | 0.3 | 5.2 | 0.5 | 2.5 | 0.1 | 6.7 | **23.3**±20.8 |
| hammer-expert | 117.0 | 129.6 | 86.7 | 25.1 | 28.5 | 75.9 | 103.3 | 86.6 | 24.8 | **133.8** | 133.5±0.4 |
| relocate-cloned | -0.1 | 0.2 | -0.1 | 0.0 | 0.0 | -0.1 | 0.0 | -0.1 | 0.0 | 0.9 | **1.2**±0.7 |
| relocate-expert | 107.3 | 106.5 | 95.0 | -0.3 | 71.9 | 82.5 | 59.3 | 0.0 | 3.4 | 106.6 | **110.5**±1.5 |
| average | 64.7 | 69.3 | 54.0 | 22.0 | 35.4 | 49.2 | 54.1 | 27.8 | 18.8 | 75.0 | **79.9** |

| MuJoCo | TD3+BC | IQL | CQL | SAC-N | EDAC | MCQ | SVR | SPOT | SAC-RND | ReBRAC | PARS |
|---|---|---|---|---|---|---|---|---|---|---|---|
| halfcheetah-r | 11.0 | 13.1 | 17.5 | 28.0 | 28.4 | 28.5 | 27.2 | 23.8 | 29.0 | 29.5 | **30.4**±0.7 |
| hopper-r | 8.5 | 7.9 | 7.9 | **31.3** | 25.3 | **31.8** | **31.0** | 31.2 | 31.3 | 8.1 | 25.4±11.5 |
| walker2d-r | 1.6 | 5.4 | 5.1 | **21.7** | 16.6 | 17.0 | 2.2 | 5.3 | **21.5** | 18.4 | 21.8±0.2 |
| halfcheetah-m | 48.3 | 47.4 | 44.0 | **67.5** | 65.9 | 64.3 | 60.5 | 58.4 | **66.6** | 65.6 | 64.2±1.2 |
| hopper-m | 59.3 | 66.3 | 58.5 | 100.3 | 101.6 | 78.4 | **103.5** | 86.0 | 97.8 | 102.0 | **104.1**±0.4 |
| walker2d-m | 83.7 | 78.3 | 72.5 | 87.9 | 92.5 | 91.0 | 92.4 | 86.4 | 91.6 | 82.5 | **97.3**±2.5 |
| halfcheetah-m-r | 44.6 | 44.2 | 45.5 | **63.9** | 61.3 | 56.8 | 52.5 | 52.2 | 54.9 | 51.0 | 57.0±0.6 |
| hopper-m-r | 60.9 | 94.7 | 95.0 | 101.8 | 101.0 | 101.6 | **103.7** | 100.2 | 100.5 | 98.1 | 103.1±0.6 |
| walker2d-m-r | 81.8 | 73.9 | 77.2 | 78.7 | 87.1 | 91.3 | **95.6** | 91.6 | 88.7 | 77.3 | 95.8±1.4 |
| halfcheetah-m-e | 90.7 | 86.7 | 91.6 | **107.1** | 106.3 | 87.5 | 94.2 | 86.9 | **107.6** | 101.1 | 103.0±2.4 |
| hopper-m-e | 98.0 | 91.5 | 105.4 | 110.1 | 110.7 | 111.2 | 111.2 | 99.3 | 109.8 | 107.0 | **113.1**±0.3 |
| walker2d-m-e | 110.1 | 109.6 | 108.8 | **116.7** | 114.7 | 114.2 | 109.3 | 112.0 | 105.0 | 111.6 | 111.8±0.7 |
| average | 58.2 | 59.9 | 60.8 | 76.3 | 76.0 | 72.8 | 73.6 | 69.4 | 75.4 | 71.0 | **77.3** |

## 6.2 BASELINE COMPARISON

**Offline training.** We evaluate PARS in comparison with 10 prior SOTA baselines: TD3+BC (Fujimoto & Gu, 2021), IQL (Kostrikov et al., 2022), CQL (Kumar et al., 2020), SAC-N (An et al., 2021), EDAC (An et al., 2021), MSG (Ghasemipour et al., 2022), SPOT (Wu et al., 2022), SVR

(Mao et al., 2024), SAC-RND (Nikulin et al., 2023), and ReBRAC (Tarasov et al., 2024). The details of the baselines are described in Appendix G.2. We primarily used the official scores reported in the respective papers for each algorithm. If a score was not available for certain datasets with a specific dataset version, we either referenced scores from other papers that benchmarked these datasets or conducted our own experiments, tuning the algorithm with the hyperparameter sets recommended by each respective author.

The evaluation results are summarized in Table 1. As shown, while other algorithms often perform well in specific domains but falter in others, PARS consistently demonstrates robust performance across a diverse range of domains. For instance, SAC-N (An et al., 2021) shows strong performance in MuJoCo; however, it struggles to learn effectively in AntMaze. In contrast, ReBRAC excels in Adroit but falls behind other algorithms in both MuJoCo and AntMaze. Specifically, in the challenging AntMaze Large and Ultra datasets, PARS achieves approximately 24% and 280% performance improvements over existing baselines, respectively. These impressive results, achieved without excessive computational resources (as shown in Appendix I) and through a straightforward implementation (as shown in Appendix E), have the potential to advance the practical application of offline RL.

Table 2: PARS evaluation on the AntMaze, Adroit, and MuJoCo domains after fine-tuning with 300k online samples. We report the final normalized score averaged over five random seeds, with $\pm$ indicating the standard deviation. The corresponding performance graphs are in Appendix A.

| AntMaze | CQL[1] | SPOT | PEX | RLPD | Cal-QL | ReBRAC | PARS |
|---|---|---|---|---|---|---|---|
| antmaze-u | **99.0**±0.6 | 98.4±1.9 | 95.2±1.6 | **99.4**±0.8 | 90.1±10.8 | **99.4**±0.9 | **99.7**±0.8 |
| antmaze-u-d | 76.9±39.7 | 55.2±32.8 | 34.8±30.1 | **99.2**±1.0 | 75.2±35.0 | 97.4±2.1 | 97.8±2.1 |
| antmaze-m-p | 94.4±3.0 | 91.2±3.8 | 83.4±2.3 | 97.4±1.4 | 95.1±6.3 | 96.8±1.9 | **99.1**±1.8 |
| antmaze-m-d | 98.8±2.5 | 91.6±3.5 | 86.6±5.0 | 98.6±1.4 | 96.3±4.8 | 95.8±3.6 | **99.4**±1.1 |
| antmaze-l-p | 87.3±5.6 | 60.4±21.5 | 56.0±3.9 | 93.0±2.5 | 75.0±14.7 | 71.4±30.9 | **96.2**±3.0 |
| antmaze-l-d | 65.3±28.3 | 69.4±23.7 | 60.4±6.8 | 90.4±3.9 | 74.4±11.8 | 89.0±3.4 | **96.8**±2.7 |
| antmaze-ultra-p | 21.3±19.0 | 0.0±0.0 | 13.3±5.8 | 8.8±16.5 | 6.9±2.7 | 0.0±0.0 | **86.5**±4.4 |
| antmaze-ultra-d | 6.3±6.6 | 5.8±11.5 | 26.7±11.3 | 40.0±37.0 | 5.7±11.2 | 1.0±1.7 | **86.4**±6.5 |
| average | 68.7 | 59.0 | 57.1 | 78.4 | 64.8 | 68.9 | **95.2** |

| Adroit | CQL | Off2On | SPOT | RLPD | Cal-QL | ReBRAC | PARS |
|---|---|---|---|---|---|---|---|
| pen-cloned | -2.6±0.1 | 102.5±166.0 | 117.1±13.4 | **154.8**±11.8 | -1.6±1.6 | 134.1±7.2 | **155.4**±3.1 |
| door-cloned | -0.34±0.00 | -8.0±0.2 | 0.05±0.06 | **110.8**±6.1 | -0.34±0.0 | 53.3±35.3 | 102.1±26.8 |
| hammer-cloned | 0.24±0.03 | -7.4±0.4 | 90.2±23.2 | 139.7±5.6 | 0.21±0.08 | 114.4±10.3 | **141.5**±1.9 |
| relocate-cloned | -0.33±0.01 | -1.5±0.5 | -0.29±0.04 | 4.8±7.1 | -0.34±0.01 | 1.5±1.1 | **53.8**±7.7 |
| average | -0.8 | 21.4 | 51.8 | 102.5 | -0.5 | 75.8 | **113.2** |

| MuJoCo | CQL | Off2On | PEX | RLPD | Cal-QL | Uni-O4 | PARS |
|---|---|---|---|---|---|---|---|
| halfcheetah-r | 26.5±3.4 | 92.7±5.7 | 60.9±5.0 | 91.5±2.5 | 32.9±8.1 | 6.8±3.9 | **100.1**±2.9 |
| hopper-r | 10.0±1.5 | 95.3±9.2 | 48.5±38.9 | 90.2±19.1 | 17.7±26.0 | 12.4±1.8 | **109.7**±5.3 |
| walker2d-r | 12.4±7.9 | 27.9±2.2 | 9.8±1.6 | 87.7±14.1 | 9.4±5.6 | 5.7±0.8 | **113.9**±13.9 |
| halfcheetah-m | 78.9±1.3 | 103.3±1.4 | 70.4±2.3 | 95.5±1.5 | 77.0±2.2 | 56.6±0.8 | **107.0**±5.0 |
| hopper-m | 100.9±0.6 | 106.3±1.7 | 86.2±26.3 | 91.4±27.8 | 100.7±0.8 | 104.8±2.6 | **111.5**±0.4 |
| walker2d-m | 88.7±0.4 | 109.8±29.6 | 91.4±14.3 | 121.6±2.3 | 97.0±8.2 | 106.5±3.4 | **126.4**±2.1 |
| halfcheetah-m-r | 50.3±28.3 | 95.6±1.7 | 55.4±5.1 | 90.1±1.3 | 62.1±1.1 | 53.2±5.4 | **98.5**±1.0 |
| hopper-m-r | 103.9±1.8 | 101.7±14.8 | 95.3±7.2 | 78.9±24.5 | 101.4±2.1 | 103.4±6.6 | **107.0**±1.4 |
| walker2d-m-r | 105.4±1.8 | 120.3±9.4 | 87.2±13.6 | 119.0±2.1 | 98.4±3.3 | 115.5±2.9 | **130.1**±4.4 |
| average | 64.1 | 96.4 | 67.2 | 96.2 | 66.3 | 62.8 | **111.6** |

**Online fine-tuning.** After offline training, we conduct online fine-tuning with 300K of online samples and compare its score with 8 prior SOTA baselines: CQL (Kumar et al., 2020), Off2On (Lee et al., 2022), SPOT (Wu et al., 2022), RLPD (Ball et al., 2023), PEX (Zhang et al., 2023), Cal-QL (Nakamoto et al., 2024), Uni-O4 (LEI et al., 2024), ReBRAC (Tarasov et al., 2024). We reproduced the results using the official implementations for all baseline scores in online fine-tuning, with details provided in Appendix G.2.

The experimental results are presented in Table 2, and the corresponding performance graphs can be found in Appendix A. Observing the results, except for two datasets, PARS outperforms all baselines across all datasets. For the online phase, RLPD, which leverages an offline dataset while

---

[1]CQL can be fine-tuned with SAC (Haarnoja et al., 2018), as proposed in Nakamoto et al. (2024).

learning from scratch online without an explicit offline training phase, showed strong performance compared to other baselines. However, PARS surpassed RLPD, challenging previous findings and highlighting that online fine-tuning can be an effective framework when proper critic regularization is applied. Notably, in the cases of AntMaze ultra-play, ultra-diverse, and Adroit relocate-cloned, PARS is the only algorithm to demonstrate strong results in these scenarios. One can argue that PARS, by reducing OOD regions beyond data coverage, may limit online exploration. However, in large and challenging tasks like AntMaze Ultra, PARS's primary success lies in reducing OOD regions during offline training, preventing exploration in completely irrelevant areas. Instead, it uses the offline data as an anchor to begin exploration, thereby significantly narrowing the search space. This can be particularly effective in online fine-tuning where online interaction steps are limited.

## 6.3 COMPARING PARS WITH GOAL-CONDITIONED OFFLINE RL IN ANTMAZE

We further compare PARS against 8 goal-conditioned offline RL baselines, specifically designed for goal-reaching tasks like AntMaze. Reaching the goal in complex, long-horizon tasks poses considerable challenges (Park et al., 2024b). To tackle these challenges, various goal-based RL methods have been developed, using techniques such as goal relabeling (Yang et al., 2022; Hejna et al., 2023), hierarchical frameworks (Park et al., 2024b), or advanced architectures like transformers (Zeng et al., 2024).

We compare PARS with 8 baselines listed in Table 3: GCBC (Ghosh et al., 2021), GC-IQL (Kostrikov et al., 2022), GC-POR (Xu et al., 2022), WGCSL (Yang et al., 2022), DWSL (Hejna et al., 2023), RvS-G (Emmons et al., 2022), HIQL (Park et al., 2024b), and GCPC (Zeng et al., 2024). The details of the baselines can be found in Appendix G.2. In Table 3, most algorithms score around 30 on the ultra dataset, with only HIQL and GCPC surpassing 50. PARS notably outperforms both, showing superior results not only on the ultra dataset but also in the overall average score across the AntMaze domain. These results highlight that, even in sparse and challenging long-horizon tasks, adhering to fundamental off-policy RL methods with proper regularization can lead to superior decision-making ability without the need for specialized designs or architectures.

Table 3: Performance comparison of PARS on AntMaze with goal-conditioned offline RL baselines.

| AntMaze | GCBC | GC-IQL | GC-POR | WGCSL | DWSL | RvS-G | HIQL | GCPC | PARS |
|---------|------|--------|--------|-------|------|-------|------|------|------|
| antmaze-u | 59.2 | 91.6 | 81.7 | 90.8 | 71.2 | 70.4 | 79.2 | 71.2 | **93.8**±2.1 |
| antmaze-u-d | 62.3 | 88.8 | 72.1 | 55.6 | 74.6 | 66.2 | 86.2 | 71.2 | **89.9**±7.5 |
| antmaze-m-p | 71.9 | 82.6 | 71.4 | 63.2 | 77.6 | 71.8 | 84.1 | 70.8 | **91.2**±3.9 |
| antmaze-m-d | 67.3 | 76.2 | 74.8 | 46.0 | 74.8 | 72.0 | 86.8 | 72.2 | **92.0**±2.2 |
| antmaze-l-p | 23.1 | 40.0 | 63.2 | 0.6 | 15.2 | 35.6 | **86.1** | 78.2 | 84.8±5.9 |
| antmaze-l-d | 20.2 | 29.8 | 49.0 | 2.4 | 19.0 | 25.2 | **88.2** | 80.6 | 83.2±5.6 |
| antmaze-u-p | 20.7 | 20.6 | 31.0 | 0.2 | 25.2 | 25.6 | 39.2 | 56.6 | **66.4**±4.4 |
| antmaze-u-d | 14.4 | 28.4 | 29.8 | 0 | 25.0 | 26.4 | 52.9 | **54.6** | 51.4±11.6 |
| average | 42.4 | 57.3 | 59.1 | 32.4 | 47.8 | 49.2 | 75.3 | 69.4 | **81.6** |

## 6.4 DISCUSSION ON THE COMPONENTS OF PARS

**How does each component of PARS affect offline performance?** To identify the source of the performance gain of PARS, we assess the offline performance across different scenarios: the critic network without LN or PA (None), with only LN (LN), and with both LN and PA (LN & PA), while varying the reward scale $c_{\text{reward}}$. The results are presented in Figure 7. As illustrated, the presence of LN leads to a trend of improved performance as $c_{\text{reward}}$ increases. Moreover, the combination with PA resulted in even higher performance, consistent with the prior discussion in Section 4.1. Additional results for component ablation can be found in Appendix H.

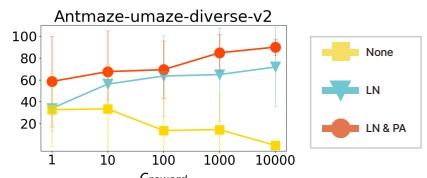

Figure 7: PARS offline performance, averaged over five seeds, with varying $c_{\text{reward}}$ and the application of LN and PA. The error bars represent the standard deviation.

**How significant is the infeasible action penalty in online fine-tuning?**

Penalizing infeasible actions (PA) can be applied not only during offline training but also in the online fine-tuning phase. In addition to its effectiveness in the offline phase, we conducted additional experiments to explore the benefits of PA in online fine-tuning by varying the application of PA during the fine-tuning phase. The results, displayed in Figure 8, demonstrate that PA enables more stable learning and accelerates performance improvement during online fine-tuning.

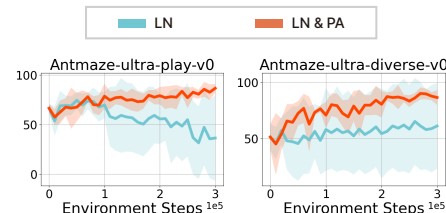

Figure 8: Ablation on PA: normalized score over 300k online fine-tuning steps.

**How far should infeasible actions be from the feasible action region?** As described in Section 4.2.3, we sample infeasible actions from the region defined by Eq. 4, which is set at a distance of the guard interval from the feasible region. The distance is determined by $|L_I|$ and $|U_I|$, and for the benchmark comparison, we set $|L_I| = |U_I|$. Figure 9 demonstrates the impact of this configuration. When $|L_I| = |U_I|$ is small, meaning the sampled infeasible actions are near the feasible region, suboptimal performance is noted, indicating that penalizing infeasible actions may influence policy evaluation within the feasible region.

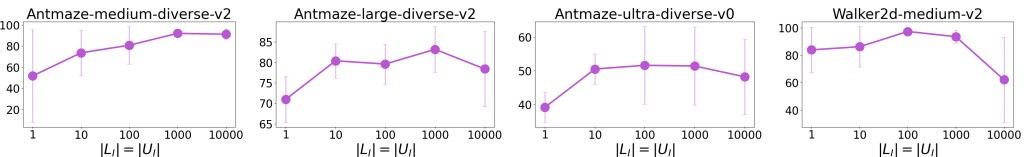

Figure 9: Final normalized score of PARS averaged over five random seeds with varying $|L_I| = |U_I|$. The error bars represent the standard deviation.

# 7 CONCLUSION

We introduce PARS, aimed at preventing critic extrapolation error and enhancing overall performance in both offline training and online fine-tuning of RL with offline data. Our analysis of LN and reward scaling with offline data reveals that combining LN with reward scaling is highly effective for mitigating OOD issues. Additionally, applying penalties to the infeasible action region reduces OOD Q-values below ID Q-values, particularly where data coverage ends, resulting in notable performance gains. PARS has demonstrated substantial performance improvements over previous SOTA algorithms across diverse RL tasks in both offline training and online fine-tuning phases. Furthermore, PARS was the only algorithm among off-policy RL variants to achieve strong performance on the notoriously challenging AntMaze Ultra task. Our findings suggest a new avenue—differing from conventional views—that strong performance across a wide array of RL tasks is achievable with only simple adjustments to off-policy algorithms, provided appropriate regularization for OOD mitigation is applied.

**Limitations and Future research.** Although PARS has achieved superior performance, it still benefits from the support of a critic ensemble, particularly in MuJoCo and Adroit. We expect that further investigation into data fitting challenges in limited datasets, coupled with potential refinements, could lead to robust performance with just a double-critic setup. As a future research direction, exploring the relationship between RS-LN and neural network expressivity would be valuable. Additionally, designing activation functions in combination with RS-LN or PA presents an intriguing research area.

## REPRODUCIBILITY STATEMENT

In Appendix E, we provide a JAX-based reference implementation of the two components of the PARS algorithm, RS-LN and PA. Furthermore, layer normalization can be seamlessly integrated into the Q network. Additionally, Appendix G contains detailed experiments and implementation settings. To further promote reproducibility, the complete training code will be publicly released on GitHub after publication.

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

# Appendices

# Contents

# A PERFORMANCE GRAPHS FOR ONLINE FINE-TUNING (300K)

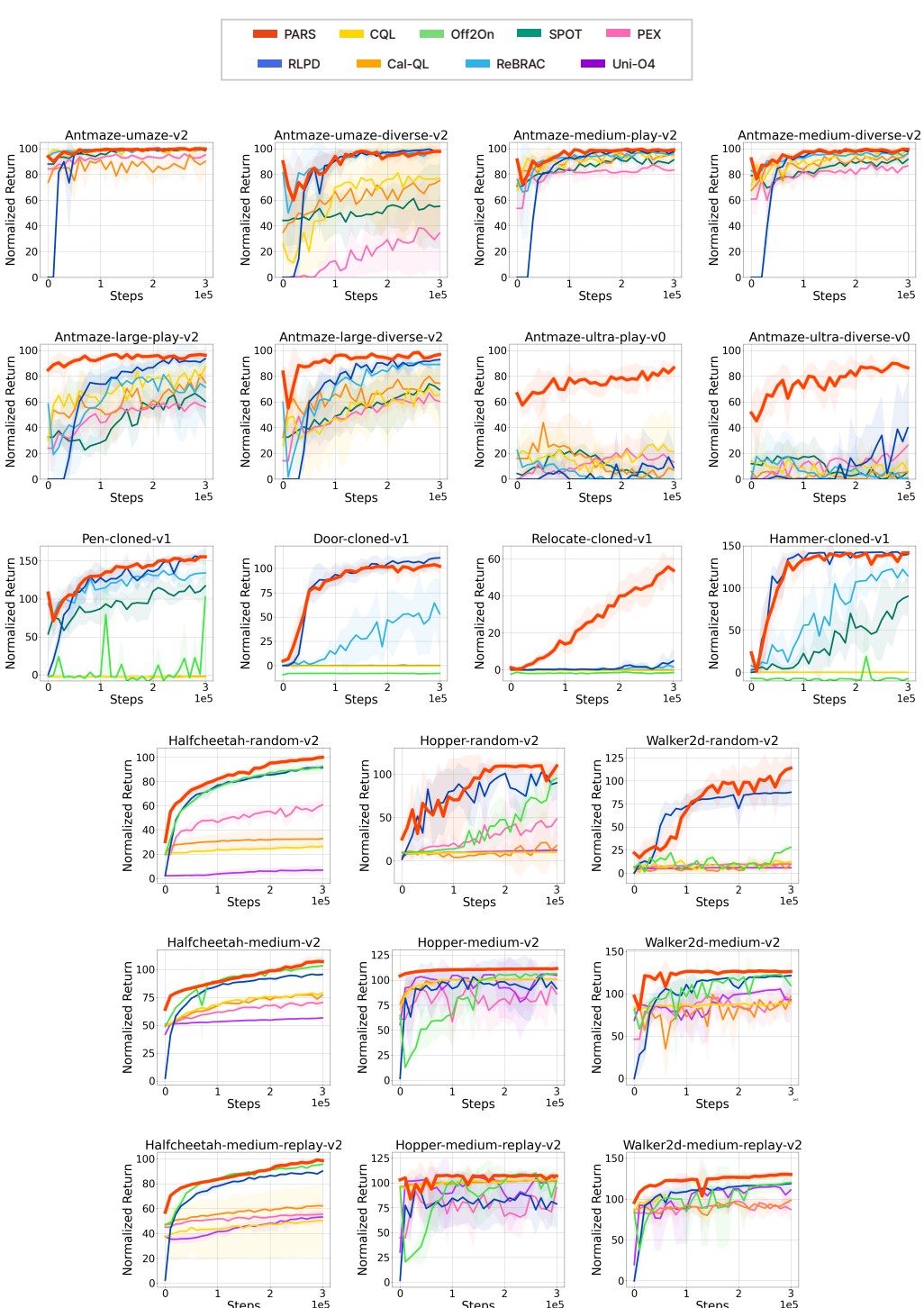

Figure 10: The performance graph of the online fine-tuning (300k), using five random seeds, corresponds to Table 2. The solid line indicates the mean, while the shaded region represents the standard deviation.

# B IN-DEPTH COMPARION WITH PRIOR WORKS

In addition to Section 3, we further examined the advantages of PARS over prior methods from the perspective of online fine-tuning. To achieve rapid progress toward the optimal policy through fine-tuning of a Q-function and policy trained on offline data with a limited number of online samples, two key factors are crucial: (1) exploring promising but unvisited regions, and (2) effectively adapting to newly collected samples.

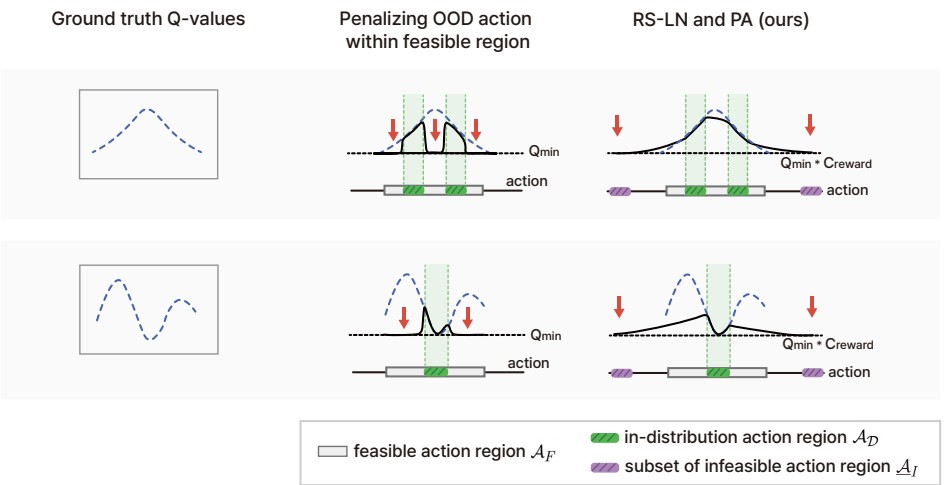

Figure 11: Comparison of the method for penalizing OOD actions in the feasible region with RS-LN and PA across various ground truth Q-values.

Figure 11 illustrates that methods penalizing OOD action regions within $\mathcal{A}_F$ tend to assign lower Q-values compared to ID regions (green areas), thereby hindering exploration during the online fine-tuning phase. In contrast, RS-LN and PA enable the network to naturally interpolate within OOD regions covered by the data, while penalizing distant infeasible regions in extrapolated areas beyond the data coverage. This approach ensures a smooth transition in Q-values between ID and OOD regions, encouraging exploration into high-value regions.

## B.1 EXPERIMENT ON Q-FUNCTION ADAPTATION

To more specifically analyze the advantages of PARS in comparison to individual prior methods during online fine-tuning, we conducted an additional experiment using a simple MDP with a single state $s$, and an action range of $[-1, 1]$. In this setup, the ground truth Q-values, $Q^*(s, a)$, are equal to the reward function, $r(s, a)$, and temporal-difference (TD) loss is defined as: $\min_\phi \mathbb{E}_{s,a\sim\mathcal{D}}\left[(Q_\phi(s,a) - Q^*(s,a))^2\right]$.

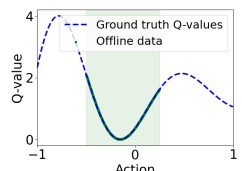

Figure 12: Ground truth Q-values with highlighted offline data region

In Figure 12, the blue dashed line represents the ground truth Q-values, while the green area denotes the samples available as offline data. We initially trained the Q-function on this offline data using prior methods, including CQL (Kumar et al., 2020), Cal-QL (Nakamoto et al., 2024), and SVR (Mao et al., 2024), as well as our proposed PARS. Subsequently, we evaluated how each Q-function adapted when new online samples were introduced.

We train the Q-function, $Q_\phi$, and the policy function, $\pi_\theta$, using a 3-layer MDP with ReLU activation. $Q_\phi$ for each algorithm was trained as follows on the offline data $\mathcal{D}$:

- CQL - We train CQL using the following policy evaluation loss function:

$$\min_\phi \mathbb{E}_{s,a\sim D}\left[(Q_\phi(s,a) - Q^*(s,a))^2\right] + \alpha\left(\mathbb{E}_{s\sim D,a\sim\pi_\theta(\cdot|s)}[Q_\phi(s,a)] - \mathbb{E}_{s,a\sim D}[Q_\phi(s,a)]\right).$$

- Cal-QL - We train Cal-QL using the following policy evaluation loss function:

$$\min_{\phi} \mathbb{E}_{s,a\sim D}\left[(Q_\phi(s,a) - Q^*(s,a))^2\right]$$
$$+ \alpha\left(\mathbb{E}_{s\sim D,a\sim\pi_\theta(\cdot|s)}[\max(Q_\phi(s,a), V^\mu(s))] - \mathbb{E}_{s,a\sim D}[Q_\phi(s,a)]\right),$$

where $V^\mu(s)$ is calculated as the mean of the $Q^*(s,\cdot)$ within the offline data distribution.

- SVR - We train SVR using the following policy evaluation operator:

$$T_{\text{SVR}}^\pi Q_\phi(s,a) = \begin{cases} T^\pi Q_\phi(s,a), & \beta(a|s) > 0, \\ Q_{\min}, & \text{otherwise.} \end{cases}$$

SVR pretrains a behavior model for $\beta(a|s)$, but we assume the true distribution is given.

- PARS - LN was applied to the Q-network, the ground truth Q-values $Q^*(s,a)$ were scaled by a factor of 100, and a $Q_{\min}$ penalty was imposed on the action ranges of (-200, -100) and (100, 200).

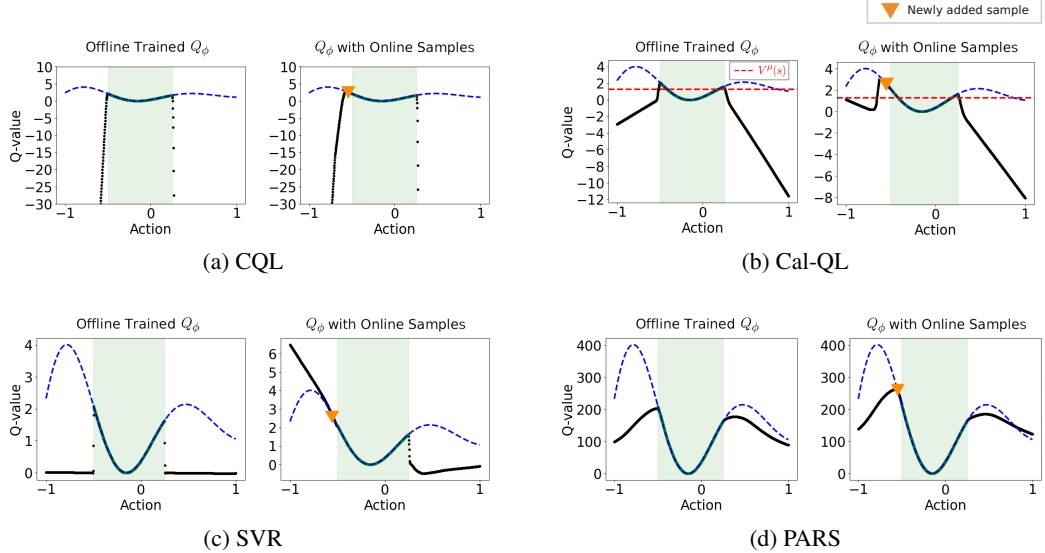

Figure 13: Comparison of how $Q_\phi$, trained on offline data with prior methods and PARS, evolves as new samples are added. For SVR, since SVR does not consider online finetuning, we do not account for the behavior model when new samples are added, focusing on the Q-function's adaptability.

Figure 13 presents the training results of $Q_\phi$ using offline data (left) and additional online samples (right, denoted by an inverted orange triangle). As shown in the figure, CQL tends to make Q-values overly conservative because its loss function minimizes the Q-values of OOD actions, which in turn slows the Q-function's adaptation to new samples. Cal-QL significantly improves upon CQL by employing a regularization technique that allows the predicted Q-value to decrease only when it is higher than $V^\mu(s)$. However, if the Q-value within the data distribution exceeds $V^\mu(s)$, particularly near the boundaries where data coverage ends, the loss minimizing the Q-value at these edges can still lead to excessive underestimation of the Q-value.

On the other hand, during SVR offline pretraining, the Q-value of OOD actions is fitted to a specific value, $Q_{\min}$. As a result, the neural network behaves unpredictably when a new sample is added. While we currently assume a single state, TD learning involves bootstrapping, and such unpredictable movements of the Q-function in one state can propagate to other states. Ultimately, this can disrupt the entire online fine-tuning process.

Unlike other methods, PARS neither explicitly minimizes the Q-value within the feasible region nor fits it to a predetermined value. This approach allows for greater flexibility in responding to new samples while leveraging the optimal actions identified during offline training as anchors for exploration. Consequently, PARS demonstrates the ability to adapt rapidly to online scenarios.

### B.2 MORE RELATED WORKS

In addition to the related works covered in Section 5, we provide a more detailed comparison with works that share commonalities with PARS.

**RLPD (Ball et al., 2023)** RLPD is an algorithm that utilizes LN in the Q network to achieve high sample efficiency in online RL with offline data, notably succeeding in sparse tasks like Antmaze. Unlike offline-to-online algorithms, RLPD does not include an offline pretraining phase. Consequently, RLPD does not explicitly account for the offline phase and, on its own, does not perform well offline.

Given our goal of achieving robust performance both offline and during the online fine-tuning phase, we adopted the LN method from RLPD but enhanced it by incorporating reward scaling and infeasible action penalties in addition to LN. This improvement effectively enhances OOD mitigation, which was insufficient with LN alone, as seen in the 'LN' column with $c_{\text{reward}} = 1$ in Figure 3.

Consequently, as shown in the performance comparison table (Table 2) and performance graph (Figure 10), PARS outperforms RLPD in more challenging tasks, such as Antmaze Ultra and Adroit, because it effectively leverages information learned during the offline phase in the online phase.

**Uni-O4 (LEI et al., 2024)** Uni-O4 is an algorithm designed to facilitate a seamless transition from offline to online learning. It achieves this by eliminating excessive conservatism or regularization during the offline training process. It shares a similar overarching goal with PARS, which aims to smoothly reduce the Q-values of OOD actions without imposing extensive conservatism.

For this purpose, Uni-O4 uses an on-policy PPO-based approach for policy learning without Q-function. It starts with a policy ensemble to capture behavior diversity and employs offline policy evaluation (OPE) to update the policy only if it outperforms the current one. Gradual updates with PPO loss and multi-step optimization ensure stability, while a unified objective integrates offline and online learning. However, if the policy ensemble lacks diversity, the initial policy may be limited, affecting updates. Moreover, OPE uses an additionally learned Transition Model and Monte Carlo methods to estimate rewards, but this introduces uncertainty that could impact performance.

Therefore, when comparing performance (Table 2 and Figure 10), PARS shows significantly superior performance compared to Uni-O4. For smooth transition to online fine-tuning, instead of focusing on multi-step policy optimization using OPE, PARS can be more efficient by utilizing a simple TD3+BC-based approach while ensuring a smooth reduction in OOD values.

## C MORE DISCUSSION ON DIDACTIC EXAMPLE

**PA without LN.** In addition to the results in Figure 3, we trained the same input $(x_1, x_2)$ using an MLP with only PA applied, without LN, and present the outcomes in Figure 14. These findings indicate that penalizing infeasible actions can mitigate overestimation in areas where data coverage ends, with its effect increasing as the reward scale $c_{\text{reward}}$ grows. However, the learned Q-values remain uneven and exhibit inaccuracies in certain regions. As demonstrated in Figure 3, the combination of PA with LN and reward scaling yields the most significant impact.

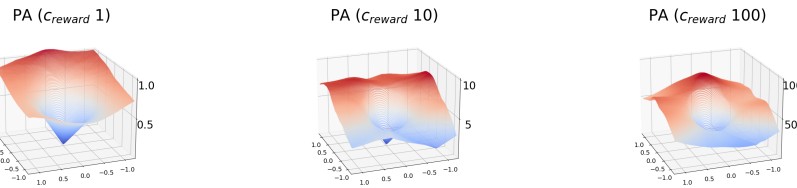

Figure 14: Results of training on a toy dataset in Section 4.1 using MLP networks with infeasible action penalty (PA) without LN, varying the $c_{\text{reward}}$.

**Analyzing split data distributions.** Figure 3 illustrates a scenario where data is assumed to be concentrated in a single region without separation. In practice, however, offline data can often be

distributed across multiple regions. To explore this, we developed a new toy dataset consisting of two slightly separated inverted cones and analyzed the learning characteristics within this dataset.

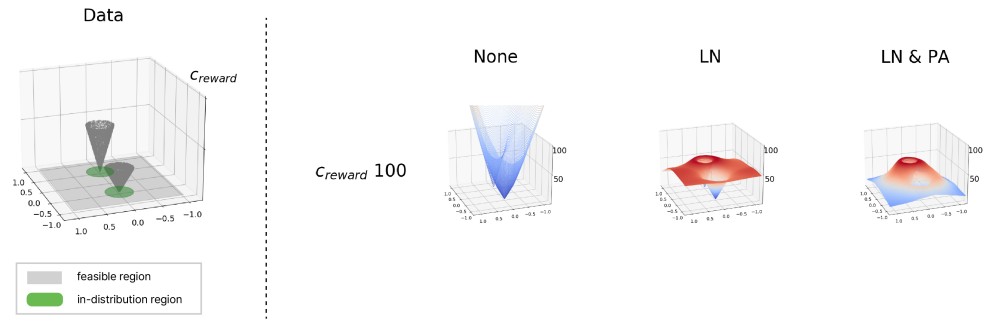

Figure 15: Results of fitting the two inverted cone-shaped input datasets using an MLP network, with and without LN and PA, at a $c_{\text{reward}}$ of 100.

Figure 15 demonstrates the data fitting behavior when $c_{\text{reward}}$ is set to 100, comparing cases where neither LN nor PA is applied ('None' column), where only LN is applied ('LN' column), and where both LN and PA are applied ('LN & PA' column). As observed, without LN or PA, the data shows significant overestimation starting from where the coverage ends, similar to the results in Figure 3. Introducing LN mitigates this overestimation, and when both LN and PA are applied, the Q-value is predicted to be lower than in the ID region. Furthermore, LN enables smooth interpolation between the two split data regions.

**Impact of activation functions.**    In Section 4.2.1, we demonstrated that RS-LN's effectiveness is tied to the network expressivity, which is influenced by the activation function (Raghu et al., 2017). The activation function plays a crucial role in determining whether neurons are activated or remain inactive during training. Accordingly, we examined how fitting characteristics, including OOD mitigation, vary depending on the activation function. We tested the toy dataset from Figure 3 in a setting with $c_{\text{reward}} = 100$ and LN applied, using GELU (Hendrycks & Gimpel, 2016), Sigmoid, SiLU (Elfwing et al., 2018), as well as cases with no activation function. The results can be seen in Figure 16. Sigmoid and SiLU showed minimal effect on mitigating Q-values, while GELU offered some benefits but was overly conservative. In the absence of activation, OOD Q-values remained high, even with a large $c_{\text{reward}}$. The impact of different activation functions extends beyond the didactic example and affects performance in real RL tasks. We elaborate on the influence of activation functions in Appendix H.

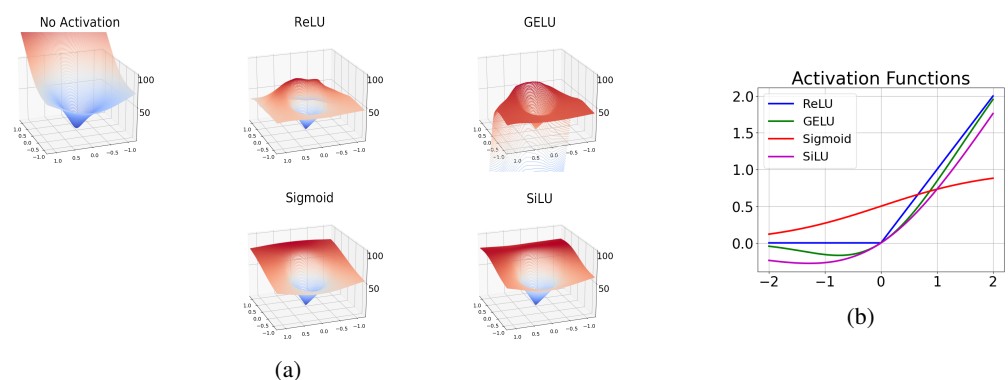

Figure 16: (a) The fitting results of applying various activation functions to the toy dataset from Figure 3 in a setting with $c_{\text{reward}} = 100$ and LN applied, (b) Plot of various activation functions.

# D IN-DEPTH ANALYSIS OF RS-LN EFFECTIVENESS

## D.1 EXPLORING THE POTENTIAL REASONS BEHIND RS-LN'S EFFECTIVENESS

In addition to the didactic example in Section 4.1 and the analysis in Figure 4, we conducted further analysis on the potential reasons why RS-LN is effective by using a Q-network consisting of three hidden layers (each composed of a linear layer, LN, and ReLU activation), trained on the Antmaze-ultra-diverse dataset. LN incorporates an affine transformation along with normalization to a mean of 0 and a variance of 1. This affine transformation mitigates the potential loss of representational capacity caused by standardizing the input distribution, thereby enhancing the model's flexibility in learning (Ba et al., 2016; Xu et al., 2019). For a hidden output $x_i$ in the $i$-th layer with a hidden dimension $d$, LN is expressed as:

$$\hat{x}_i = \frac{x_i - \mu}{\sigma}\eta + \beta,$$

where $\eta$ and $\beta$ are learnable parameters with a dimension of $d$. Here, $\eta$ controls the scale of the normalized values, while $\beta$ adjusts their offset. Starting from an initial value of 1, $\eta$ is trained to naturally increase in response to the growth of $c_{\text{reward}}$, effectively adapting to the change.

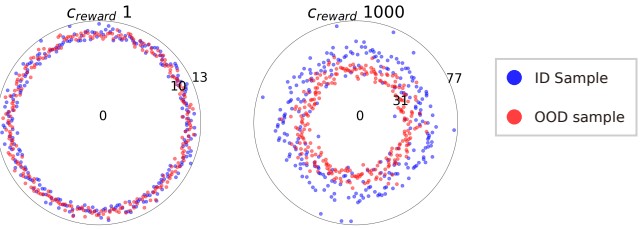

Figure 17: A visualization of the norm of the LN output in the final layer, $\|\hat{x}_{-1}\|$, for ID/OOD samples. The radius of the circle represents the norm value. ID samples are obtained by sampling from the dataset, while OOD samples are generated by sampling actions from $\mathcal{A}_I$ for the same states sampled from the dataset.

Using the Q-network with varying $c_{\text{reward}}$, as shown in Figure 17, we observed during the joint optimization of the LN parameter $\eta$ and the linear layer parameters that, as $c_{\text{reward}}$ increases, the norm of the LN output in the final layer, $\|\hat{x}_{-1}\|$, becomes more distinguishable. Specifically, it exhibits higher values for ID samples and lower values for OOD samples. Notably, in Figure 17, for $c_{\text{reward}} = 1000$ (right), while ID and OOD samples are distinguishable, the gap between them does not widen abruptly but instead increases gradually. This observation aligns with the explanation in Appendix B.1, where PARS naturally reduces the Q-values of OOD samples, resulting in significant benefits for online fine-tuning.

However, this observation requires further analysis and represents one of the important future directions. A deeper investigation into layer normalization and its interaction with reward scaling would be an intriguing research avenue, offering opportunities for further exploration and advancements.

## D.2 COMPARISON OF TD3+BC AND IQL CRITIC UPDATES AND THE EFFECTIVENESS OF RS-LN

Here, we compare the critic update of TD3+BC (Fujimoto & Gu, 2021), as used in PARS, with the critic updates of in-sample learning-based methods, focusing on a representative method, IQL (Kostrikov et al., 2022), to analyze in depth whether RS-LN can be broadly applied.

First, the TD3+BC critic update loss function is defined as:

$$\mathcal{L}_Q = \min_\phi \mathbb{E}_{s,a,s'\sim\mathcal{D}}\left[\left(Q_\phi(s,a) - \left(c_{\text{reward}}\cdot r(s,a) + \gamma\mathbb{E}_{a'\sim\pi_\theta(\cdot|s')}Q_\phi(s',a')\right)\right)^2\right]. \quad (7)$$

Additionally, the critic update loss function of IQL is expressed as:

$$\mathcal{L}_Q = \min_\phi \mathbb{E}_{s,a,s'\sim\mathcal{D}}\left[\left(Q_\phi(s,a) - (c_{\text{reward}}\cdot r(s,a) + \gamma V_\psi(s'))\right)^2\right], \quad (8)$$

with an additional value function loss that aligns $V_\psi(s)$ with $Q_\phi(s, a)$:

$$\mathcal{L}_V = \min_\psi \mathbb{E}_{s,a\sim\mathcal{D}} \left[ \mathcal{L}_\tau^2 \left( Q_\phi(s, a) - V_\psi(s) \right) \right], \tag{9}$$

where expectile regression is defined as $\mathcal{L}_\eta^2(u) = |\eta - \mathbb{1}(u < 0)|u^2$, with $\eta \in [0.5, 1)$, to formulate the asymmetrical loss function for the value network $V_\psi$.

Therefore, the critic updates in TD3+BC and IQL can be expressed in a unified form. The generalized critic loss is as follows:

$$\mathcal{L}_{\text{Critic}} = \min_\phi \mathbb{E}_{s,a,s'\sim\mathcal{D}} \left[ (Q_\phi(s, a) - Y(s, a, s'))^2 \right], \tag{10}$$

where the target $Y(s, a, s')$ is defined as:

$$Y(s, a, s') = c_{\text{reward}} \cdot r(s, a) + \gamma G(s'). \tag{11}$$

The key difference lies in the definition of the next-state value estimate $G(s')$:

$$G(s') = \begin{cases} \mathbb{E}_{a'\sim\pi_\theta(\cdot|s')}Q_\phi(s', a'), & \text{(TD3+BC)} \\ V_\psi(s'), & \text{(IQL)} \end{cases}. \tag{12}$$

Although the target $Y(s, a, s')$ varies between TD3+BC and IQL and is dynamically adjusted throughout the learning process rather than remaining fixed, the core task of fitting $Q_\phi$ to $Y$ follows a structurally similar approach in both methods. Each method minimizes the squared error between $Q_\phi$ and the changing $Y$, providing a common framework for understanding their update mechanisms. In this context, both methods share the characteristic that an increase in $c_{\text{reward}}$ leads to an increase in the target value $Y$ and that the application of LN helps mitigate catastrophic overestimation. As analyzed in Figures 3 and 4, this can have the same effect on regression fitting and network expressivity, suggesting the potential for the general applicability of RS-LN.

# E   REFERENCE IMPLEMENTATION

We provide a JAX-based reference implementation for the critic loss, and the complete training code will be made publicly available on GitHub following publication.

```
def _critic_loss(self,
        critic_params,
        critic_target_params,
        actor_target_params,
        transition,
        rng):
    state, action, next_state, reward, not_done = transition

    # Scale the reward by the predefined reward scaling factor
    reward = self.reward_scale * reward

    # Compute the target Q-value (implementation details omitted)
    target_Q = ...  # Placeholder for target Q-value computation

    # Get current Q estimates (implementation details omitted)
    current_Q = ...  # Placeholder for current Q estimate computation

    # Compute critic loss
    Q_loss = jnp.mean(jnp.square(current_Q - target_Q[:, 0][None, ...]))

    # Uniform infeasible action sampling
    infeasible_action = (jax.random.uniform(rng, action.shape) * 2) - 1
    infeasible_action = jnp.where(infeasible_action < 0, infeasible_action - 1,
        infeasible_action + 1) * self.L

    # Compute current Q-value for infeasible action
    current_Q_infeasible = self.critic_model.apply(critic_params, state, infeasible_action)

    # Set target Q-value for infeasible action to min_q
    target_Q_infeasible = jnp.ones_like(target_Q) * self.min_q

    # Compute infeasible Q loss
```

```
Q_loss_infeasible = jnp.mean(jnp.square(current_Q_infeasible - target_Q_infeasible[:, 0][
    None, ...]))

# Combine the losses
critic_loss = Q_loss + self.alpha * Q_loss_infeasible

return critic_loss
```

Listing 1: An example of reward scaling and infeasible action sampling implementation in JAX.

## F    EXTENSION TO OTHER OFFLINE RL METHODS

We also investigate whether incorporating RS-LN and PA into other existing offline RL algorithms leads to performance gains.

### F.1    EXTENSION TO SAC-BASED METHODS

Since PARS is based on TD3+BC, we applied RS-LN and PA to SAC-RND (Nikulin et al., 2023), an offline variant of SAC (Haarnoja et al., 2018). SAC-RND already uses a reward scaling of 100 with LN, and we explored whether further increasing it, along with applying PA, would enhance performance. For hyperparameters, we used SAC-RND's actor and critic beta values from Nikulin et al. (2023) and tuned PARS's $\alpha$ and $c_{\text{reward}}$ to 0.001, 0.0001, and 1000, 10000, respectively. Table 4 shows that applying PA and reward scaling leads to a noticeable performance improvement when integrated into an offline SAC-based algorithm. Although this study focuses on implementing our proposed PARS with TD3+BC, it can be seamlessly extended to other off-policy algorithms and their applications, demonstrating its versatility and effectiveness.

Table 4: Performance improvement after applying PA and RS to SAC-RND. The scores are the averages of the final evaluations across five random seeds.

| AntMaze | SAC-RND (Tarasov et al., 2024) | SAC-RND (reproduce) | SAC-RND with PA and RS-LN (Ours) |
|---|---|---|---|
| umaze-diverse-v2 | 66.0±25.0 | 69.7±26.4 | **83.0**±14.3 |
| medium-play-v2 | 38.5±29.4 | 52.4±21.6 | **79.2**±5.9 |
| medium-diverse-v2 | 74.7±10.7 | 75.3±11.2 | **84.7**±8.3 |
| large-play-v2 | 43.9±29.2 | 40.0±25.8 | **53.9**±12.1 |
| large-diverse-v2 | 45.7±28.5 | 37.2±25.4 | **54.2**±14.7 |

### F.2    EXTENSION TO IN-SAMPLE LEARNING-BASED METHODS

Both TD3+BC and SAC are actor-critic methods that may sample OOD actions during training. Alternatively, in-sample learning-based methods (Kostrikov et al., 2022; Xu et al., 2023; Garg et al., 2023; Hansen-Estruch et al., 2023) avoid OOD action sampling by implicitly learning the maximum Q-values, focusing on the upper portion of the Q-value distribution using only in-sample data. Building on the previous discussion, we investigated whether PA and RS-LN could be applied to these methods. Specifically, we conducted experiments with IQL (Kostrikov et al., 2022), a representative in-sample learning method, setting the discount factor $\gamma$ to 0.995 and increasing the temperature parameter $\tau$ to 20. Consistently, we also varied the $\gamma$ and $\tau$ in IQL and reported the best results. We used a $c_{\text{reward}}$ value of 1000 for RS-LN.

Table 5: Performance improvement after applying PA and RS-LN to IQL. The scores are the averages of the final evaluations across five random seeds.

| AntMaze | IQL (Kostrikov et al., 2022) | IQL (reproduce) | IQL with RS-LN (Ours) | IQL with PA and RS-LN (Ours) |
|---|---|---|---|---|
| umaze-diverse-v2 | 62.2 | 66.5±5.5 | **83.0**±6.2 | 81.1±7.9 |
| large-play-v2 | 39.6 | 45.4±5.8 | **60.4**±6.2 | **60.6**±5.3 |
| ultra-play-v2 | - | 13.3±5.7 | **36.6**±13.3 | **37.3**±10.7 |

As shown in Table 5, PA was not particularly effective in this setting, likely due to its nature as a method that avoids OOD sampling. Conversely, RS-LN demonstrated noticeable effectiveness, indicating its potential for broader applicability and effectiveness when integrated into in-sample learning-based approaches. However, it did not surpass the performance of TD3+BC-based PARS. This can be attributed to the following reasons.

First, IQL relies on AWR (Peng et al., 2019) for policy extraction. As highlighted by Park et al. (2024a), this approach has inherent limitations because it exclusively depends on a mode-covering behavioral cloning term, which restricts actions to remain within the convex hull of dataset actions. Additionally, as discussed in Hansen-Estruch et al. (2023), methods such as expectiles, quantiles, and exponentials can be used to implicitly learn maximum Q-values, each with its unique characteristics and limitations. Thus, the choice of implicit Q-learning method tends to impose certain constraints. Consequently, the best performance was achieved when PA and RS-LN were applied to a fundamental off-policy algorithm, such as TD3, with further explanation of the effectiveness of their combination in Appendix K. Exploring the extension of PA and RS-LN to in-sample learning, with variations in critic loss function design and policy extraction methods, presents an interesting direction for future research.

## G EXPERIMENT DETAILS

### G.1 BENCHMARK DETAILS

Fu et al., 2020 introduced a variety of datasets designed for different RL tasks, such as AntMaze, Adroit, and MuJoCo. Additionally, zhengyao jiang et al., 2023 expanded the AntMaze domain by proposing an ultra dataset, featuring a larger map size than previously proposed, thus increasing the complexity of the task.

For the AntMaze domain, we leverage eight datasets: {umaze-v2, umaze-diverse-v2, medium-play-v2, medium-diverse-v2, large-play-v2, large-diverse-v2, ultra-play-v0, ultra-diverse-v0}. These datasets encompass different levels of difficulty based on the maze's size, complexity, and the diversity of start and goal positions.

In the Adroit domain, we utilize four tasks: {pen, door, hammer, relocate}, each associated with two dataset qualities: {cloned-v1, expert-v1}. The cloned datasets are created by training an imitation policy using demonstration data, executing the policy, and mixing the resulting data with the original demonstrations in a 50-50 ratio. In contrast, expert datasets are derived from fine-tuned RL policy.

For the MuJoCo domain, we use three tasks: {halfcheetah, hopper, walker2d}, each of which has four dataset qualities: {random-v2, medium-replay-v2, medium-v2, medium-expert-v2}. The random dataset consists of data generated by randomly initialized policies, the medium dataset comes from partially trained policies, the medium-replay dataset includes data from replay buffers during training, and the expert dataset contains demonstrations from well-trained agents performing near-optimal behavior.

### G.2 BASELINES

**Offline training.** We primarily relied on the officially reported scores from each paper for datasets benchmarked for comparison. For datasets that were not benchmarked in the original paper, we referred to scores from other works that reported results for those datasets. For datasets without available scores from other sources, we reproduced the results using the respective implementations, tuning hyperparameters according to the recommendations in each paper.

Specifically, for datasets not benchmarked in the original paper, we either obtained the scores or conducted experiments as outlined below. In all other cases, we used the scores reported in the original paper. In cases where we conducted the experiments, we reported the final evaluation scores using five random seeds, with the mean and standard deviation provided in Tables 6 and 7.

- AntMaze: To begin, since the AntMaze Ultra datasets had not been benchmarked in previous studies we compared, we conducted experiments for all prior baselines. For MSG and SAC-RND, we ran the experiments using the official implementations. For TD3+BC, IQL, CQL, SPOT, and ReBRAC, we used the CORL library (Tarasov et al., 2022), which provides a single-file implemen-

tation of state-of-the-art (SOTA) offline RL algorithms. For the remaining datasets, we referenced the SAC-N and EDAC scores from Tarasov et al. (2022). Additionally, we sourced the MSG and SAC-RND scores from Tarasov et al. (2024), as those works benchmarked the v0 and v1 versions of the AntMaze datasets, rather than the v2 versions.

- Adroit: For Adroit, we obtained the TD3+BC, IQL, CQL, and SAC-RND scores from Tarasov et al. (2024), and the SAC-RND and EDAC scores for expert datasets from Tarasov et al. (2022). Moreover, for SPOT and SVR, we ran the experiments using the CORL library and the official SVR implementation, respectively.

- MuJoCo: For MuJoCo, we obtained the IQL and CQL scores for random datasets from Lyu et al. (2022). Moreover, we ran SPOT on random datasets using the CORL library.

The URLs for each implementation are listed below:

- CORL - `https://github.com/tinkoff-ai/CORL`
- MSG - `https://github.com/google-research/google-research/tree/master/jrl`
- SAC-RND - `https://github.com/tinkoff-ai/sac-rnd`
- SVR - `https://github.com/MAOYIXIU/SVR`

Table 6: Final normalized evaluation scores averaged over five random seeds for SPOT and SVR. For each dataset, we tuned them as recommended in the paper and reported the best scores.

| Dataset | SPOT |
|---|---|
| halfcheetah-r | 23.8±0.5 |
| hopper-r | 31.2±0.4 |
| walker2d-r | 5.3±9.3 |
| pen-cloned | 15.2±18.7 |
| pen-expert | 117.3±14.9 |
| door-cloned | 0.0±0.0 |
| door-expert | 0.2±0.0 |
| hammer-cloned | 2.5±3.2 |
| hammer-expert | 86.6±46.3 |
| relocate-cloned | -0.1±0.0 |
| relocate-expert | 0.0±0.0 |

| Dataset | SVR |
|---|---|
| pen-cloned | 65.6±18.8 |
| pen-expert | 119.9±11.2 |
| door-cloned | 1.1±1.6 |
| door-expert | 83.3±14.9 |
| hammer-cloned | 0.5±0.4 |
| hammer-expert | 103.3±16.2 |
| relocate-cloned | 0.0±0.0 |
| relocate-expert | 59.3±10.2 |

| Dataset | MCQ |
|---|---|
| antmaze-u | 27.5±20.6 |
| antmaze-u-d | 0.0±0.0 |
| antmaze-m-p | 0.0±0.0 |
| antmaze-m-d | 0.0±0.0 |
| antmaze-l-p | 0.0±0.0 |
| antmaze-l-d | 0.0±0.0 |
| pen-cloned | 35.3±28.1 |
| pen-expert | 121.2±15.9 |
| door-cloned | 0.2±0.5 |
| door-expert | 73.0±2.2 |
| hammer-cloned | 5.2±6.3 |
| hammer-expert | 75.9±30.2 |
| relocate-cloned | -0.1±0.0 |
| relocate-expert | 82.5±7.2 |

Table 7: Final normalized evaluation scores averaged over five random seeds for the baselines on the AntMaze Ultra datasets. For each dataset, we tuned them as recommended in the paper and reported the best scores.

| Dataset | TD3+BC | IQL | CQL | MCQ | MSG | SPOT | SAC-RND | ReBRAC |
|---|---|---|---|---|---|---|---|---|
| antmaze-ultra-p | 0.0±0.0 | 13.3±5.7 | 16.1±8.5 | 0.0±0.0 | 0.6±0.9 | 4.4±1.3 | 20.6±15.0 | 22.4±11.7 |
| antmaze-ultra-d | 0.0±0.0 | 14.2±6.2 | 6.5±3.5 | 0.0±0.0 | 1.0±1.4 | 12.0±4.4 | 10.5±8.8 | 0.8±1.8 |

**Online fine-tuning.** During the online fine-tuning phase, since the official score for 300k online samples is typically unavailable, we re-run all the baselines using their corresponding official implementations, except for CQL, which uses the provided code from Cal-QL. The URLs for each implementation are listed below:

- RLPD - `https://github.com/ikostrikov/rlpd`
- Cal-QL - `https://github.com/nakamotoo/Cal-QL`
- Off2On - `https://github.com/shlee94/Off2OnRL`
- PEX - `https://github.com/Haichao-Zhang/PEX`
- Uni-O4 - `https://github.com/Lei-Kun/Uni-O4`

- SPOT - `https://github.com/thuml/SPOT`

**Goal-conditioned offline RL.** We obtain the GCBC, GC-POR, and HIQL scores from Park et al., 2024b, and the GC-IQL, WGCSL, DWSL, RvS-G, and GCPC scores from Zeng et al., 2024. For the umaze and umaze-diverse scores for GCBC, GC-POR, and HIQL, we run the experiments using the implementations provided for each respective algorithm in the official HIQL repository (`https://github.com/seohongpark/HIQL`).

Table 8: Final normalized evaluation scores averaged over five random seeds for GCBC, GC-POR, and HIQL on the AntMaze umaze and umaze-diverse datasets.

| Dataset | GCBC | GC-POR | HIQL |
|---|---|---|---|
| antmaze-umaze | 59.2±13.3 | 81.7±9.5 | 79.2±4.2 |
| antmaze-umaze-diverse | 62.3±13.8 | 72.1±12.3 | 86.2±5.7 |

### G.3 PARS

We built our code on the JAX (Bradbury et al., 2018) implementation of TD3 (Fujimoto et al., 2018) (`https://github.com/yifan12wu/td3-jax`) and made modifications to suit the PARS algorithm, such as adding an infeasible action penalty, and reward scaling. For critic ensembles, we referenced the implementation of SAC-N (`https://github.com/Howuhh/sac-n-jax`). Our full implementation code for PARS will be released after publication.

Table 9: $Q_{\min}$ for each task.

| AntMaze | $Q_{\min}$ |
|---|---|
| antmaze-umaze | 0 |
| antmaze-medium | 0 |
| antmaze-large | 0 |
| antmaze-ultra | 0 |
| **Adroit** | $Q_{\min}$ |
| pen | -715 |
| door | -42 |
| hammer | -348 |
| relocate | 0 |
| **MuJoCo** | $Q_{\min}$ |
| halfcheetah | -366 |
| hopper | -166 |
| walker2d | -229 |

Table 10: PARS's general hyperparameters.

| Offline Hyperparameters | Value |
|---|---|
| optimizer | Adam (Kingma & Ba, 2015) |
| batch size | 256 |
| learning rate (all networks) | 3e-4 |
| tau ($\tau$) | 5e-3 |
| hidden dim (all networks) | 256 |
| gamma ($\gamma$) | 0.995 on AntMaze, 0.99 on others |
| infeasible region distance | 1000 on AntMaze, 100 on others |
| actor cosine scheduling | True on Adroit, False on others |
| nonlinearity | ReLU (Agarap, 2018) |
| **Online Fine-Tuning Hyperparameters** | **Value** |
| exploration noise | 0.1 on MuJoCo 0.05 on others |
| learning starts | 0 |
| update to data (UTD) ratio | 20 |

**AntMaze.** For offline training, we tuned $c_{\text{reward}}$ to 100, 1000, and 10,000, $\beta$ between 0.005 and 0.01, and $\alpha$ between 0.001 and 0.01. During online fine-tuning, we used a 50/50 mix of offline and online data. The online $\beta$ was tuned to 0, 0.001, and 0.01, while $\alpha$ was fixed at 0.001, except for ultra-play, which used 0.0001, and $S_{k_{\text{actor}}}$ was set to 1. The hyperparameters for each dataset are listed in Table 11.

**Adroit.** For offline training, we set $c_{\text{reward}}$ to 10 and tuned $\beta$ between 0.1 and 0.01, and $\alpha$ to 0.001, 0.01, and 0.1. During online fine-tuning, a 50/50 mix of offline and online data was used. The online $\alpha$ was set to 0.001, and $\beta$ was tuned to 0 and 0.01, and $S_{k_{\text{actor}}}$ was set to 1. The specific hyperparameters for each dataset are provided in Table 11.

**MuJoCo.** For offline training, we used a $c_{\text{reward}}$ of 5 for HalfCheetah and a $c_{\text{reward}}$ of 10 for Walker2d and Hopper. The $\beta$ was set to 0, and $\alpha$ was tuned between 0.01, 0.001 and 0.0001. Additionally, for this domain, we found that adjusting policy noise and $S_{k_{\text{critic}}}$ provides further benefits, so we varied the policy noise between 0 and 0.2, and $S_{k_{\text{critic}}}$ between 2 and 10. During online fine-tuning, we used 5% offline data for the HalfCheetah and random datasets, and 50% for the remaining datasets. We also tuned $\alpha$ among 0.1, 0.01, and 0.0001, and $S_{k_{\text{actor}}}$ between 1 and 10. The specific hyperparameters for each dataset can be found in Table 12.

Table 11: Dataset-specific hyperparameters of PARS for AntMaze and Adroit domains used in offline training and online fine-tuning.

| | Offline | | | Online | | | Offline | | Online |
|---|---|---|---|---|---|---|---|---|---|
| **AntMaze** | $c_{\text{reward}}$ | $\beta$ | $\alpha$ | $\beta$ | **Adroit** | $\beta$ | $\alpha$ | $\beta$ |
| antmaze-umaze | 10000 | 0.005 | 0.001 | 0 | pen-cloned | 0.01 | 0.01 | 0 |
| antmaze-umaze-diverse | 10000 | 0.005 | 0.001 | 0.001 | door-cloned | 0.01 | 0.01 | 0.01 |
| antmaze-medium-play | 1000 | 0.01 | 0.001 | 0 | hammer-cloned | 0.1 | 0.001 | 0 |
| antmaze-medium-diverse | 1000 | 0.01 | 0.001 | 0 | relocate-cloned | 0.01 | 0.01 | 0.01 |
| antmaze-large-play | 1000 | 0.01 | 0.001 | 0.01 | pen-expert | 0.01 | 0.01 | - |
| antmaze-large-diverse | 10000 | 0.01 | 0.01 | 0.01 | door-expert | 0.1 | 0.001 | - |
| antmaze-ultra-play | 100 | 0.01 | 0.001 | 0.001 | hammer-expert | 0.01 | 0.001 | - |
| antmaze-ultra-diverse | 10000 | 0.01 | 0.01 | 0.01 | relocate-expert | 0.1 | 0.0001 | - |

Table 12: Dataset-specific hyperparameters of PARS for MuJoCo domain used in offline training and online fine-tuning.

| | Offline | | | Online | |
|---|---|---|---|---|---|
| **MuJoCo** | $\alpha$ | policy noise | $\mathcal{S}_{k_{\text{critic}}}$ | $\alpha$ | $\mathcal{S}_{k_{\text{actor}}}$ |
| halfcheetah-random | 0.0001 | 0.2 | 2 | 0.0001 | 1 |
| halfcheetah-medium | 0.0001 | 0 | 2 | 0.0001 | 10 |
| halfcheetah-medium-replay | 0.0001 | 0 | 2 | 0.0001 | 10 |
| halfcheetah-medium-expert | 0.0001 | 0.2 | 10 | - | - |
| hopper-random | 0.01 | 0.2 | 2 | 0.01 | 1 |
| hopper-medium | 0.01 | 0 | 10 | 0.1 | 1 |
| hopper-medium-replay | 0.01 | 0 | 10 | 0.1 | 1 |
| hopper-medium-expert | 0.0001 | 0.2 | 10 | - | - |
| walker2d-random | 0.01 | 0 | 10 | 0.0001 | 10 |
| walker2d-medium | 0.01 | 0 | 10 | 0.1 | 1 |
| walker2d-medium-replay | 0.01 | 0 | 10 | 0.01 | 1 |
| walker2d-medium-expert | 0.0001 | 0.2 | 10 | - | - |

## H  MORE ABLATION STUDY

**Extended results on the impact of PARS components on offline performance.**    In addition to Figure 7 in Section 6.4, we conducted additional experiments on various datasets to further explore the impact of the PARS components in offline training. As shown in Figure 18, beyond AntMaze, the application of RS-LN in MuJoCo and Adroit leads to a general improvement in performance. Furthermore, incorporating PA results in a more robust enhancement.

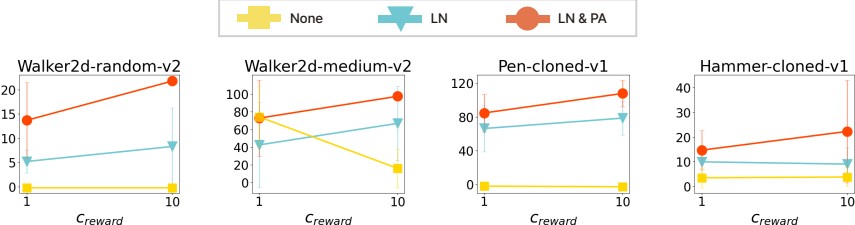

Figure 18: Extended ablation results of PARS components beyond Figure 7. We evaluate PARS with varying $c_{\text{reward}}$ and the application of LN and PA, averaged over five random seeds. The error bars represent the standard deviation.

**Impact of the number of critic ensembles.**    As discussed in Section 4.2.3, PARS can be used in combination with a critic ensemble. We analyzed the effect of the number of critic ensembles on offline performance in Figure 19. As shown in the figure, while the impact is minimal in the Antmaze domain, incorporating an ensemble in the MuJoCo and Adroit domains enables more stable learning and thus contributes to improved performance.

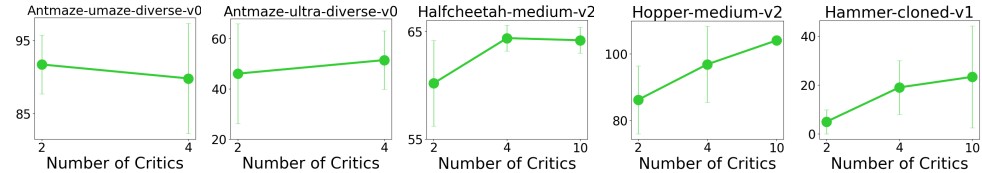

Figure 19: Final normalized score of PARS for offline training, averaged over five random seeds with varying numbers of critics. Error bars represent the standard deviation.

**Which activation function would be most compatible with PARS?**    In Appendix C, we examine the effect of activation functions on fitting toy data. Additionally, we analyze how activation functions, when combined with PARS, influence performance in real RL tasks.

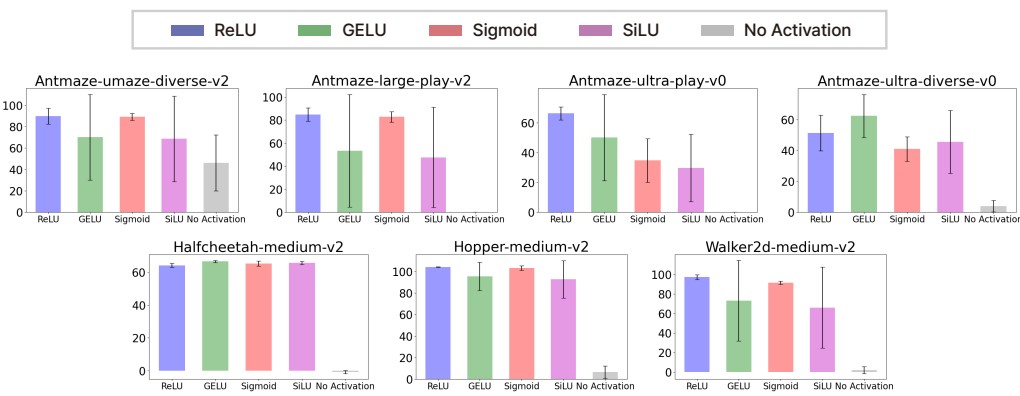

Figure 20: Final normalized score of PARS for offline training averaged over five random seeds with varying activation functions. The error bars represent the standard deviation.

Figure 20 presents the results of applying each activation function to PARS, showing that the ReLU activation function consistently performs well across various tasks. While other activation functions outperform ReLU in some tasks, they lack robustness across all tasks. The impact of activation functions on RL tasks, in conjunction with RS-LN and PA, could be an interesting topic for future research.

# I COMPUTATION COST

We compared the training time and GPU memory usage of PARS with various offline baselines. The comparison was conducted using a single L40S GPU, and the training time was measured over 5000 gradient steps. The baselines were implemented using either PyTorch or JAX. Given that JAX is generally recognized for its speed advantage over PyTorch due to optimizations like just-in-time compilation and efficient hardware utilization (Bradbury et al., 2018), we distinguished the training time and GPU memory usage for PyTorch and JAX with yellow and blue bars, respectively. PARS, implemented in JAX, is indicated by a red bar.

Showing the comparison results presented in Figure 21, PARS has faster training time compared to methods like SAC-N (PyTorch) and MSG (JAX), which use a large number of critic ensembles. Additionally, while SAC-RND and ReBRAC have faster training times than PARS, they use significantly more GPU memory. In contrast, PARS efficiently reduces computation costs by using both less training time and less GPU memory.

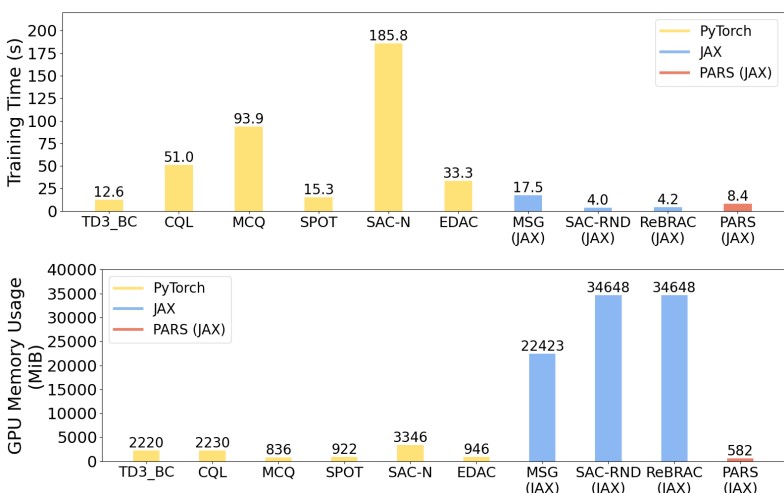

Figure 21: Comparison of PARS's training time and GPU memory usage with various offline baselines. Yellow bars represent PyTorch implementations, blue bars represent JAX implementations, and the red bar represents PARS implemented in JAX.

## J  DIDACTIC EXAMPLE WITH MORE SEEDS AND INCREASED $c_{\text{REWARD}}$

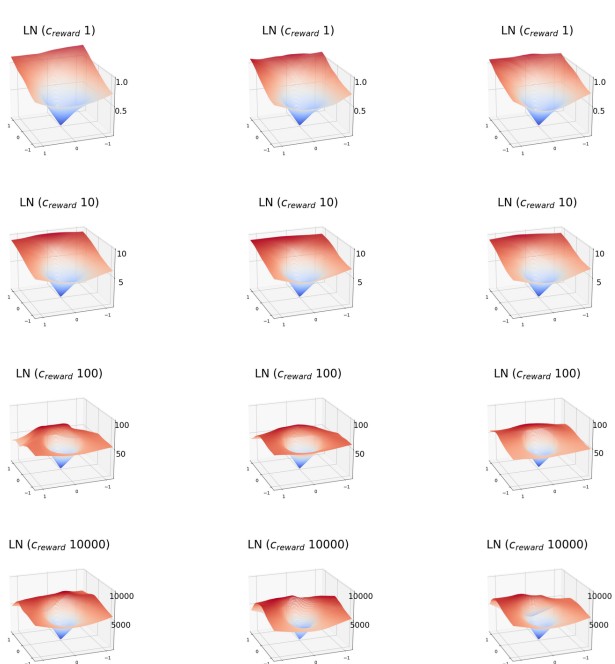

Figure 22: LN applied with three seeds (corresponding to the 'LN' column in Figure 3)

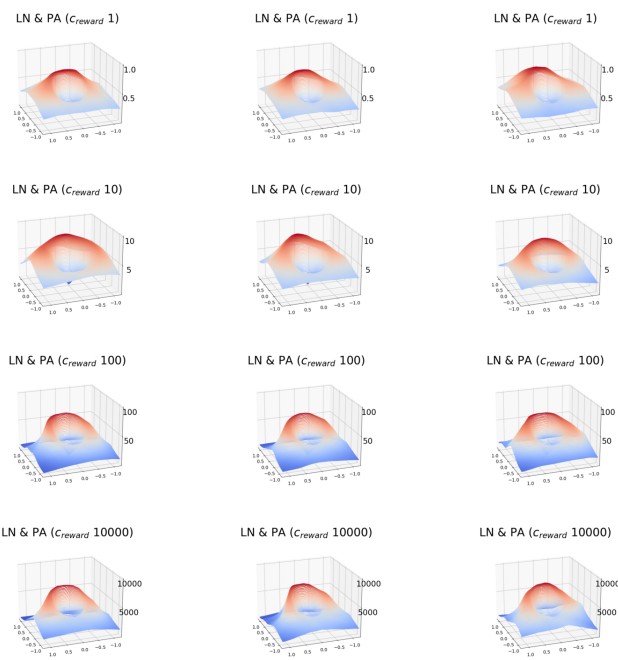

Figure 23: LN and PA applied with three seeds (corresponding to the 'LN & PA' column in Figure 3)

## K   WHY IS THE COMBINATION OF TD3+BC WITH RS-LN AND PA EFFECTIVE?

As discussed in the Experiments section (Section 6), PARS achieves high performance based on TD3+BC. Notably, as mentioned in Appendix F.2, TD3+BC demonstrated more pronounced effects compared to in-sample learning-based methods, such as IQL, that utilize weighted behavior cloning for policy extraction (Kostrikov et al., 2022; Xu et al., 2023). Why does the combination of TD3+BC with the components of PARS—RS-LN and PA—produce better results? This can be understood by examining the characteristics of TD3+BC alongside the features of PARS.

Following the discussion in Park et al. (2024a), we first compare two popular policy extraction objectives: behavior-constrained policy gradient (e.g., DDPG+BC (Park et al., 2024a), TD3+BC) and weighted behavior cloning (e.g., AWR (Peng et al., 2019), IQL). We refer the reader to Park et al. (2024a) for a more in-depth comparison of policy extraction methods.

**(1) Behavior-constrained policy gradient (e.g., DDPG+BC, TD3+BC)**

$$\max_{\pi} \mathcal{J}_{\text{TD3+BC}}(\pi) = \mathbb{E}_{s,a\sim\mathcal{D}}\left[Q(s,\pi(s)) - \alpha(\pi(s) - a)^2\right],$$

with $\alpha$ to control the strength of the BC regularizer.

**(2) Weighted behavioral cloning (e.g., AWR, IQL)**

$$\max_{\pi} \mathcal{J}_{\text{IQL}}(\pi) = \mathbb{E}_{s,a\sim\mathcal{D}}\left[e^{\alpha(Q(s,a)-V(s))}\log\pi(a\mid s)\right],$$

with $\alpha$ to control the (inverse) temperature.

As highlighted in Park et al. (2024a), TD3+BC, a behavior-constrained policy gradient algorithm, employs both mode-seeking first-order critic maximization and mode-covering behavioral cloning. This mechanism enables controlled extrapolation by "hillclimb" the critic function while maintaining proximity to the mode. Therefore, as illustrated in Figure 24, while methods like weighted behavioral cloning predominantly generate actions within $\mathcal{A}_{\mathcal{D}}$, TD3+BC allows actions to be generated beyond the boundaries of $\mathcal{A}_{\mathcal{D}}$ to some extent.

Although this flexibility can make TD3+BC more unstable than weighted behavioral cloning when the critic's values for OOD actions are highly inaccurate, it can offer greater advantages if the critic is more accurate and generalizes well. As discussed in Section 3 and Appendix B, PARS smoothly reduces the values of OOD actions outside the data coverage while seamlessly interpolating the values of OOD actions within the data coverage. Therefore, PARS effectively prevents overestimation while maintaining smoothness, thereby unlocking significant synergy with TD3+BC.

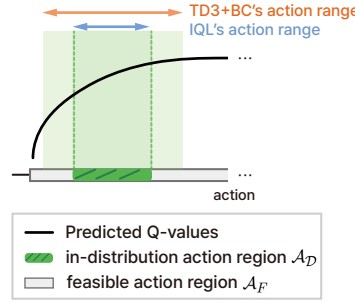

Figure 24: Comparison of the range of actions that can be generated by IQL and TD3+BC for the same predicted Q-values. TD3+BC enables the generation of actions over a slightly broader range than $\mathcal{A}_{\mathcal{D}}$.

In summary, TD3+BC can achieve greater synergy when the critic's predictions are both accurate and well-generalized. The superior performance of PARS within TD3+BC can be largely attributed to the role of PARS's RS-LN and PA in fostering a critic with high accuracy and improved generalization capabilities.

