# OpenReview forum: "Penalizing Infeasible Actions and Reward Scaling in Reinforcement Learning with Offline Data"
_ICLR.cc/2025/Conference — Submitted to ICLR 2025_

### Official Review · Reviewer_h6w1 · 2024-10-21

**Soundness:** 3
**Presentation:** 3
**Contribution:** 2
**Rating:** 6
**Confidence:** 2

**Summary:**

The authors present a concise yet effective algorithm called PARS, which introduces Reward Scaling combined with Layer Normalization (RS-LN) to mitigate extrapolation errors and enhance the offline and offline-to-online performance. The **experimental results** are particularly **impressive**.

However, I have some questions: 1) How does PARS differ from other related algorithms? Such as [1] 2) Could the authors clarify how the algorithm's design specifically contributes to its performance advantage? (see question section)

If you can given clarification, I think this paper will be much more comprehensive.

**Strengths:**

The advantages of PARS are primarily reflected in two key aspects:

- PARS is a straightforward approach that integrates Layer Normalization and reward scaling with TD3+BC, making it easy to implement.

- PARS demonstrates strong performance across both offline and offline-to-online tasks from a variety of domains, including Androids, Mujoco, and Antmaze, making it highly scalable and showcasing competitive experimental results.

**Weaknesses:**

The experimental performance of PARS is impressive. However, the authors do not thoroughly discuss the relationship between the performance improvements and the algorithm's design. Additionally, it is necessary to provide clearer distinctions between PARS and similar algorithms, such as [1], as mentioned in the questions section. Specifically, please see the **question section**.

**Questions:**

**Q.1** Please clarify the advantages of your research's LN compared to that in [1] ? And could you demonstrate your methods' advantage over other offline-to-online algorithms? Such as [1], Cal-QL, RLPD, PEX.

**Q.2** Subsequently, please clarify why your algorithm performs better in online fine-tuning.

Cal-QL [2] suggests that value misalignment in both offline and online settings may be a reason for poor performance during online fine-tuning. If you chose to utilize layer-normalization, it implies that your algorithm may learn more conservative Q-values, potentially leading to a more conservative performance during online fine-tuning. However, the algorithm in [1] improves upon soft-actor-critic (SAC) during online fine-tuning, where maximizing entropy encourages the agent to explore more diverse samples, thus alleviating the conservatism inherited from the offline stages.

Meanwhile, I can give you some related studies for reference. Recent research has shown that appropriately incorporating BC terms during the online process may enhance online performance [3]. Could you please give much more analytical explanation such as [3], at the very least about the potential reasons, thank you.

**Q.3** Please clarify the innovations of your algorithm compared to [1]. Specifically, is the improvement in your algorithm mainly due to the combined use of reward re-scaling and layer normalization (LN), or is it due to the combined use of TD3-BC, and the reward scaling algorithm? Thank you.

*If TD3+BC is not the majority reason for PARS' high efficiency, please answer **Q.4***, otherwise, you shouldn't have to answer **Q.4**

**Q.4** Could you please conduct additional experiments to verify the extensibility of PARS. For instance, you could replace TD3+BC with other methods, especially algorithms beyond policy constraints, such as CQL/IQL etc.

# Reference

[1]Philip J. Ball, Laura Smith, Ilya Kostrikov, Sergey Levine, Efficient Online Reinforcement Learning with Offline Data.

[2] Mitsuhiko Nakamoto, Yuexiang Zhai, Anikait Singh, Max Sobol Mark, Yi Ma, Chelsea Finn, Aviral Kumar, Sergey Levine, Cal-QL: Calibrated Offline RL Pre-Training for Efficient Online Fine-Tuning

[3] Seohong Park, Kevin Frans, Sergey Levine, Aviral Kumar. Is Value Learning Really the Main Bottleneck in Offline RL?

---

> ### Author Response · Authors · 2024-11-21
> **Authors' response to reviewer h6w1**
>
> We are thankful for the encouraging feedback on PARS's effectiveness and for the chance to further highlight its strengths in comparison to previous offline-to-online works, as well as to better articulate the underlying factors contributing to its effectiveness.
>
> &nbsp;
>
> ### **[W1] Relationship between the performance improvements and the algorithm's design.**
>
> We would appreciate it if the reviewer could check our analysis presented in Figure 7, Figure 8, and Appendix H (Figure 18). We have already conducted an ablation study on various components of PARS. Figures 7 and 18 examine how LN, $c_\text{reward}$, and PA impact performance during the offline phase, while Figure 8 further analyzes the effects of PA during the online fine-tuning phase. The analysis shows that without LN, performance decreases as $c_\text{reward}$ increases, but with LN, performance improves as $c_\text{reward}$ grows. Additionally, when PA is incorporated, the performance becomes even stronger and more robust.
>
> &nbsp;
>
> ### **[W2 & Q1] Comparison with RLPD**
>
> We want to clarify that RLPD, reference [1] mentioned by the reviewer, is not an offline-to-online algorithm because it does not include an offline pretraining phase. Instead, it leverages offline data to make online learning more sample-efficient. While it shares a similar goal with offline-to-online algorithms—achieving high online performance with minimal samples—its approach is different. Offline-to-online algorithms aim to adapt well to distribution shifts and effectively utilize offline knowledge. In contrast, RLPD learns from scratch without explicit offline pretraining while utilizing offline data to achieve sample-efficient learning.
>
> Therefore, RLPD does not explicitly consider the offline phase, and as shown in Table R5, RLPD alone does not perform well offline. **Since our goal is to achieve robust performance both offline and during the online fine-tuning phase, we applied the method of using layer normalization from RLPD but improved it by incorporating reward scaling and infeasible action penalties in addition to layer normalization. This improvement effectively enhances OOD mitigation, which was insufficient with layer normalization alone, as observed in the 'LN' column with $c_\text{reward}$ 1 in Figure 3.** Consequently, as demonstrated in Table 2 and Figure 10 of our manuscript, PARS outperforms RLPD in more challenging environments like Antmaze Ultra and Adroit because it effectively utilizes the information learned during the offline phase in the online phase.
>
> **[Table R5]** Comparison of the offline learning results using the RLPD algorithm using five random seeds with the PARS offline results.
>
> |  | RLPD offline | PARS offline |
> | --- | --- | --- |
> | hopper-medium | 4.41±1.2 | 104.1 |
> | walker2d-medium | 0.22±0.5 | 97.3 |
> | halfcheetah-medium | 59.6±6.0 | 64.2 |
> | antmaze-umaze | 0.0±0.0 | 93.8 |
> | antmaze-medium-play | 0.0±0.0 | 91.2 |
> | antmaze-large-play | 0.0±0.0 | 84.8 |
>
> &nbsp;
>
> ### **[Q1 & Q2] Comparison with other offline-to-online algorithms**
>
> **In the revised paper's Appendix B, Figure 13, we added a comparison experiment to highlight the benefits of PARS over prior offline-to-online algorithms, including Cal-QL.** Cal-QL builds on CQL by addressing its limitation of being overly conservative when predicting Q-values for OOD actions, which hinders online fine-tuning. It achieves this improvement by learning calibrated conservative value functions. While Cal-QL makes notable advancements over CQL through allowing the predicted Q-value to decrease only when it exceeds $V^{\mu}(s)$, challenges remain. Specifically, if the Q-value within the data distribution surpasses $V^{\mu}(s)$, particularly near the boundaries where data coverage ends, minimizing the Q-value at these edges may still result in significant underestimation.
>
> On the other hand, PARS, which naturally reduces the Q values of OOD actions through RS-LN and PA, benefits during online fine-tuning by promoting exploration within the potential region and effectively adapting to new samples. While the reviewer mentioned that LN could be conservative, LN distinguishes itself from conservative loss function-based methods by allowing Q values to naturally decrease, offering a unique advantage.
>
> In addition to the offline-to-online algorithms discussed in Appendix B, PEX integrates offline-trained policies into online learning through a policy expansion strategy that preserves offline behaviors while enabling further learning. This strategy freezes the offline policy to prevent degradation and introduces a new learnable policy, allowing both to collaborate adaptively. PARS focuses on critic regularization, whereas PEX emphasizes policy and can be considered an orthogonal line of work. The performance comparison in Table 2 shows that PARS significantly outperforms PEX.

---

> ### Author Response · Authors · 2024-11-21
> **Authors' response to reviewer h6w1**
>
> ### **[Q3] Origin of PARS’s innovation**
>
> Thank you for the important question! PARS achieves good performance by effectively combining the two sources mentioned by the reviewer. Looking into each of them in more detail:
>
> 1. Reward scaling and layer normalization (RS-LN)
>
>     As mentioned in our response to [W1], the analysis experiments on the effects of reward re-scaling and layer normalization can be found in Figures 7 and 18 of the manuscript. The results indicate that when these two components are combined, they have a significantly positive impact on performance. **Notably, the combination of reward scaling with LN demonstrates significantly enhanced performance compared to the LN used in RLPD. Additionally, incorporating PA into this setup further amplifies the performance gains.**
>
> 2. The combined use of TD3+BC
>
>     Moreover, according to [Ref.13], DDPG+BC is more effective than AWR [Ref.9] in achieving optimal performance offline. Citing the explanation from [Ref.13]: *“AWR only has a mode-covering weighted behavioral cloning term, while DDPG+BC has both mode-seeking first-order value maximization and mode-covering behavioral cloning terms. As a result, actions learned by AWR always lie within the convex hull of dataset actions, whereas DDPG+BC can ‘hillclimb’ the learned value function, even allowing extrapolation to some degree while not deviating too far from the mode.”*
>
>     As a variant within the same framework as DDPG+BC, TD3+BC similarly enjoys comparable advantages. In our [Q4] response, we present a more detailed analysis of the outcomes when the PARS method is applied to IQL, which incorporates AWR. The findings reveal that while IQL+PARS demonstrates noticeable improvements, TD3+BC, which integrates both “mode-seeking” and “mode-covering”, proves to be more effective in achieving superior performance.
>
> &nbsp;
>
> ### **[Q4] Regarding extensibility of PARS**
>
> We previously provided experiments in Appendix F demonstrating the application of the PARS idea to SAC-RND, which showed improved performance compared to the original SAC-RND.
>
> Furthermore, we extended the PARS idea to in-sample learning-based methods such as IQL. As shown in Appendix F.2, since IQL does not perform OOD sampling, PA does not provide benefits for this method. Conversely, RS-LN demonstrated noticeable effectiveness, highlighting its potential for broader applicability and impact when integrated into in-sample learning-based approaches. However, it did not surpass the performance of TD3+BC-based PARS, as discussed in our response to [Q3] and detailed in Appendix F.2.
>
> &nbsp;
>
> [Ref.9] Peng, Xue Bin, et al. "Advantage-weighted regression: Simple and scalable off-policy reinforcement learning." *arXiv* 2019.
>
> [Ref.13] Seohong Park, Kevin Frans, Sergey Levine, Aviral Kumar. Is Value Learning Really the Main Bottleneck in Offline RL?

---

> > ### Comment · Reviewer_h6w1 · 2024-11-22
> > **Official comments by Reviewer h6w1**
> >
> > Thanks for your efforts in responding to the questions. I will follow up on a few questions.
> >
> > **Regarding W2 && Q1.** I think this part of the response makes sense.
> >
> > **Regarding Q1 && Q2.** I believe the author's paper is already relatively comprehensive to a certain extent. However, I welcome the author to continue adding more comparisons with past different offline-to-online algorithms, either intuitively or theoretically, in the appendix. Alternatively, summarizing similar methods would also be welcome. **This will not affect my rating**. I suggest this because there is a wealth of literature on the field of offline-to-online methods, and Cal-QL and PEX are just two examples among many.
> >
> > **Regarding Q3.** Could you please add the connection between the mode seeking and mode covering you analyzed and your proposed solution to the appendix or the main text? I believe this point will help readers understand why RS-LN, when combined with TD3-BC, can achieve such good experimental results. Thank you.
> >
> > **Regarding Q4.** Thank you for your response. Could you further analyze whether RS-LN is theoretically a universal algorithm in the context of in-sample learning-based approaches? (**I'm not requesting additional experiments, or proofs, so the absence of them will not affect my rating negatively.**) If it is, could you briefly mention the potential reasons for this method's scalability? Could this information be added in the paper or on an appendix page? Thank you.

---

> ### Author Response · Authors · 2024-11-24
>
> Thank you for the additional comments, and we're glad to hear that our response regarding RLPD was helpful. We've included answers to the reviewer's additional questions in our revised paper as follows:
>
> - **Regarding Q1 & Q2. → Appendix B.2**
>
>     We previously compared offline-to-online algorithms in Section 5, Related Works, and added a more detailed explanation of two works, RLPD and Uni-O4, which we believe are particularly related to PARS, in Appendix B.2. If there are any works that the reviewer think should be compared but we haven't, please let us know!
>
> - **Regarding Q3. → Appendix K**
>
>     We additionally analyzed in Appendix K why RS-LN and PA are more effective when combined with TD3+BC. The policy extraction methods differ between TD3+BC and IQL: TD3+BC employs a behavior-constrained policy gradient, while IQL relies on weighted behavioral cloning. Specifically, TD3+BC employs both mode-seeking first-order critic maximization and mode-covering behavioral cloning. Consequently, as illustrated in Figure 24 of our revised paper, TD3+BC enables the generation of actions that extend beyond the boundaries of $\mathcal{A}_\mathcal{D}$ to some extent. This approach can achieve greater synergy when the critic's predictions are both accurate and well-generalized. The superior performance of PARS within TD3+BC can largely be attributed to the role of RS-LN and PA in fostering a critic with high accuracy and improved generalization capabilities. For more details, please refer to Appendix K.
>     - We added this at the end of the appendix to avoid disrupting section numbering, which could confuse reviewers when referencing section numbers to read our previous responses. As the reviewer noted, these explanations enhance the paper's comprehensiveness, so we're considering moving them to the front of the appendix or main text.
>
> - **Regarding Q4. → Appendix D.2**
>
>     We further analyzed how RS-LN can also be effective when combined with in-sample learning-based methods by comparing the representative IQL critic learning objective with the TD3+BC critic learning objective. For more details, please refer to Appendix D.2.
>
>
> Let us know freely if you have any additional questions or if your concern has been addressed.

---

> ### Comment · Reviewer_h6w1 · 2024-11-24
> **Futher questions**
>
> Thanks for answering my question. Regarding Q1 and Q2, I have another related question: could you please investigate any potential connections between RLPD and ENOTO[1]? ENOTO adapts Q ensemble. Q ensemble originates from SAC-N [2], and through ensemble Q, it can indirectly penalize samples that appear relatively infrequently, thereby mitigating the Out-Of-Distribution (OOD) problem. The approach adopted by ENOTO is to normally use Q ensemble during offline pre-training and gradually relax the ensemble during online fine-tuning. Thank you.
>
> # Reference
>
> [1] Kai Zhao, Jianye Hao, Yi Ma, Jinyi Liu, Yan Zheng, Zhaopeng Meng, ENOTO: Improving Offline-to-Online Reinforcement Learning with Q-Ensembles
>
> [2] Gaon An, Seungyong Moon, Jang-Hyun Kim, Hyun Oh Song, Uncertainty-Based Offline Reinforcement Learning with Diversified Q-Ensemble

---

> > ### Author Response · Authors · 2024-11-25
> >
> > Thank you for following up and introducing a recent relevant study! ENOTO focuses on leveraging ensembles for efficient offline-to-online RL and has demonstrated its applicability to various existing works that utilize ensembles in a plug-in manner. We have also added an explanation about ENOTO in Section 5 (the Related Works section).
> >
> > In addition to partially mitigating the OOD problem, as the reviewer mentioned, ENOTO also utilizes ensembles to enable efficient online exploration during fine-tuning. Both ENOTO and RLPD use critic ensembles, but they differ in how these ensembles are utilized, such as in calculating critic targets and facilitating online exploration. For example, when calculating critic targets, RLPD samples two ensembles and uses the minimum value among them, whereas ENOTO calculates the expected value of the minimum across all possible pairs of ensembles. Moreover, RLPD does not incorporate a specific exploration mechanism. In contrast, ENOTO employs ensemble-based weighted Bellman backups from SUNRISE [Ref. 14] to encourage more optimistic exploration during online fine-tuning. Furthermore, while ENOTO places greater emphasis on the methods of utilizing ensembles, RLPD combines ensembles with other design choices (e.g., LayerNorm) without exploring the ensembles themselves as extensively as ENOTO.
> >
> > Exploring how ensembles could be leveraged in online RL to further enhance sample efficiency and online exploration is indeed an intriguing future research direction! Additionally, since ENOTO can be applied to various algorithms, investigating the effectiveness of PARS+ENOTO would also be a promising avenue for future work.
> >
> > In this work, we have demonstrated the feasibility of developing a simple yet powerful RL algorithm that performs effectively in both offline and online phases, supported by various analyses of the effectiveness of RS-LN and PA. We are optimistic that combining this approach with future research on ensemble utilization methods could further enhance the performance and efficiency of the current PARS.
> >
> > We hope our response sufficiently addresses the reviewer’s question.
> >
> > &nbsp;
> >
> > [Ref.14] Lee, Kimin, et al. "Sunrise: A simple unified framework for ensemble learning in deep reinforcement learning." *ICML*, 2021.

---

> ### Comment · Reviewer_h6w1 · 2024-11-25
> **Official comments by Reviewer h6w1**
>
> Thanks for your response, which has addressed my concerns. Additionally, I believe that the experiments and evaluations conducted on RLPD are quite thorough. Therefore, I am prepared to increase the score to 6. RLPD is also a comprehensive paper, and as such, the author has successfully balanced the length and expression in their writing to some extent. Meanwhile, I need to lower my confidence from 3 to 2.

---

> > ### Author Response · Authors · 2024-11-25
> >
> > Thank you for your reply. We’re glad our response addressed your concerns, and we sincerely appreciate the time and effort you put into offering valuable feedback to enhance our work.

---

### Official Review · Reviewer_UZZ9 · 2024-11-05

**Soundness:** 3
**Presentation:** 3
**Contribution:** 2
**Rating:** 6
**Confidence:** 4

**Summary:**

This paper uses reward scaling and penalizing techniques for offline RL. The proposed method is not novel but effective. Experiments show the advantages of the proposed method.

**Strengths:**

The writing of the paper is very clear and easy to follow.

The idea of the paper makes sense and is intuitive. The idea of imposing penalizing infeasible actions is simple but effective. It does not need additional efforts to train a model to classify the OOD and ID.

The experiments of the paper are sufficient and valid.

**Weaknesses:**

The proposed method is very simple (which can also be regarded as an advantage). It is unclear to me why the layer normalization and reward scaling can mitigate OOD overestimation. The paper could elaborate more insight on this.



I appreciate the Didactic exmaple in Sec 4.1, which is good and intuitive. However, it's an example. More analysis and discussion would be good. Please see my questions below.





Missing Ablation studies:

- The paper can also compare with method that use a behavior model to classify the ID and OOD. Could the PA also show benefits over that?



Minor comments:

- Line 019: lower than what?
- Line $s_t,a_t$ should be defined before using them,.
- Line 089: the returns should be expected returns.p

**Questions:**

Regarding the Didactic example in Sec 4.1:

- The examples are all positive. What happens if the values are negative? By the way, what is the sign of the reward in the practical environments used in the paper?
- It seems from both Figure 3 and the experimental results, with LN, larger $c_{\text{reward}}$ always leads to better results. Could you elaborate on the reason? And is it correct that we can choose a larger $c$?

Would the method and the main result also work and hold for a discrete action space?

The proposed method shows large improvements on AntMaze? Could the author elaborate more on these results?

I don't really understand why $L_I = U_I$ in Line 501. Why are they equal?

---

> ### Author Response · Authors · 2024-11-21
> **Authors' response to reviewer UZZ9**
>
> We appreciate the positive comments on the simple and intuitive strengths of PARS, as well as the feedback to clarify the underlying causes and unclear points. Responding to these comments allowed us to improve the clarity and explanatory depth of our work.
>
> &nbsp;
>
> ### **[W1] Why the RS-LN can mitigate OOD overestimation.**
>
> Thank you for the important question. In Section 4.1, we illustrate through a didactic example that when reward scaling is combined with layer normalization, OOD Q-values progressively decrease further. Furthermore, the analysis in Section 4.2.1 demonstrates that this phenomenon is related to network expressivity, analyzed through factors such as dormant neurons. As highlighted in Appendix C, the ReLU activation function also plays a role.
>
> In the revised paper, Appendix D further addresses the potential reasons why RS-LN can mitigate OOD overestimation. LN not only normalizes inputs to have a mean of 0 and a variance of 1 but also adjusts the scale through learnable parameters. This scaling is trained to progressively increase from an initial value of 1 in response to reward scaling. During the joint optimization of the scaling parameter and other parameters in the Q-network, we observed a phenomenon where the feature norm of ID samples becomes more distinguishable from that of OOD samples, as shown in Figure 17. Specifically, the feature norm of ID samples is larger than that of OOD samples. While this observation requires further analysis, it could serve as a crucial topic for future work.
>
> We have identified important phenomena related to RS-LN and demonstrated its empirical effectiveness through analyses using various metrics. These findings could inspire important future research directions, and we look forward to seeing further advancements in this area.
>
> &nbsp;
>
> ### **[W2] Could PA also show benefits over methods using behavior models?**
>
> - The first advantage of PA, compared to methods that rely on behavior models, is that it eliminates the need to train an auxiliary model. Additionally, because a behavior model is a predictive model, its accuracy inevitably decreases as the data distribution becomes more complex. In such cases, the Q-values for ID actions may also be incorrectly decreased. Moreover, as noted by AWR [Ref.10] and SPOT [Ref.11], continuously tuning a behavior model in an online environment where the data distribution is constantly changing is impractical.
> - Moreover, compared to the approach of using a behavior model to distinguish between ID/OOD and lowering the OOD to a specific value, **PA demonstrates significant advantages in online fine-tuning, particularly in terms of online exploration and adapting well to new samples, as we have shown through various figures and additional experiments in Appendix B of the revised paper.**
>
> &nbsp;
>
> ### **[W3] Regarding writing**
>
> Thank you for providing comments on the writing. We have incorporated them into the revised paper.
>
> &nbsp;
>
> ### **[Q1] Regarding the sign of the rewards**
>
> In the domains we benchmarked, MuJoCo and Adroit have mixed ± rewards, while Antmaze follows a sparse reward setting where a positive reward of +1 is given upon reaching the goal. For example, the reward statistics in the MuJoCo and Adroit datasets are as follows:
>
> **[Table R4]** Reward statistics of MuJoCo and Adroit datasets.
>
> |  | min | median | max |
> | --- | --- | --- | --- |
> | hopper-random | -1.2 | 0.9 | 3.9 |
> | walker2d-random | -2.3 | 0.1 | 2.7 |
> | pen-cloned | -6.2 | 10.9 | 61.0 |
> | door-cloned | -0.5 | -0.3 | 20.0 |
>
> Therefore, PARS performs well in scenarios where rewards are either all positive or a mix of positive and negative values. However, as the reviewer pointed out, if all rewards are negative, our method cannot demonstrate its benefits. In such cases, the rewards in the offline data can be converted to positive values before applying our method.

---

> > ### Author Response · Authors · 2024-11-21
> > **Authors' response to reviewer UZZ9**
> >
> > ### **[Q2] Why does PARS perform well with a large $c_\text{reward}$, and does a larger $c_\text{reward}$ tend to yield better results?**
> >
> > In our response to [W1], the reviewer can find the explanation of how RS-LN addresses OOD overestimation. Additionally, to verify whether a larger $c_\text{reward}$ is always beneficial, we increased $c_\text{reward}$ up to $10^{10}$ and conducted additional experiments on the antmaze-umaze-diverse dataset using five random seeds. We found that stable results can be achieved with a mean performance of 97.3, even when $c_\text{reward}$ increases up to $10^{6}$. However, performance began to decrease at $c_\text{reward} = 10^{8}$, with a mean performance of 81.3 at $c_\text{reward} = 10^{8}$ and 45.4 at $c_\text{reward} = 10^{10}$. Specifically, when $c_\text{reward}$ was $10^{8}$, the loss function increased to $10^{16}$, and when $c_\text{reward}$ reached $10^{10}$, the loss further escalated to $10^{20}$. Such a large loss scale could potentially lead to overflow or gradient explosion, highlighting the importance of carefully managing the loss scale.
> >
> > &nbsp;
> >
> > ### **[Q3] Regarding application to discrete action space**
> >
> > Thank you for your valuable suggestion regarding broader research directions. CQL (Conservative Q-Learning)[Ref.12] has demonstrated its effectiveness when applied to discrete action spaces, such as the Atari environment. The CQL paper shows that a well-designed critic regularization method can also achieve notable results in discrete action spaces. Since PARS has shown significant advantages over CQL in offline and offline-to-online settings for continuous action spaces, we anticipate that it could similarly be effective when applied to discrete action spaces. However, like many recent RL works, our current study focuses primarily on continuous action spaces and does not analyze discrete action spaces in depth. Extending the ideas of PARS to discrete action spaces could indeed be a promising direction for future work.
> >
> > &nbsp;
> >
> > ### **[Q4] Why does PARS show large improvements on AntMaze?**
> >
> > As discussed in [Ref.4], LN is crucial for achieving strong performance, particularly in settings with sparse rewards, limited demonstrations, and narrow offline data coverage. Antmaze is an example of a sparse reward setting where a reward of 1 is only given upon reaching the goal, resulting in infrequent reward signals. This scarcity and imbalance in signals can exacerbate critic instability. Consequently, this environment has historically posed significant challenges, making our method—designed to amplify the scale of reward signals and fully utilize network expressivity—especially effective in addressing these issues.
> >
> > &nbsp;
> >
> > ### **[Q5] Regarding $L_i$ = $U_i$**
> >
> > Apologies for the confusion. The equation $L_i = U_i$ was a typo; it should actually be $|L_i| = |U_i|$. This has been corrected in the revised paper.
> >
> > &nbsp;
> >
> > [Ref.4] Ball, Philip J., et al. "Efficient online reinforcement learning with offline data." *ICML 2023*.
> >
> > [Ref.9] Sokar, Ghada, et al. "The dormant neuron phenomenon in deep reinforcement learning." *ICML 2023*.
> >
> > [Ref.10] Peng, Xue Bin, et al. "Advantage-weighted regression: Simple and scalable off-policy reinforcement learning." *arXiv* 2019.
> >
> > [Ref.11] Wu, Jialong, et al. "Supported policy optimization for offline reinforcement learning." *NeurIPS 2022*.
> >
> > [Ref.12] Kumar, Aviral, et al. "Conservative q-learning for offline reinforcement learning." *NeurIPS 2020*.

---

> ### Author Response · Authors · 2024-11-26
>
> Thank you once again for the reviewer's efforts in providing thoughtful comments on our work. In our new common response, we have summarized the paper revisions made during the rebuttal period. We would be delighted if our revised paper helps provide a clearer understanding of our work.
>
> In particular, we discussed [W1] why RS-LN can mitigate OOD overestimation in Appendix D, and delved deeper into the content of [W2] in Appendix B. Other responses can be found in our previous responses. Please feel free to reach out if you have any additional questions.

---

### Official Review · Reviewer_6JBM · 2024-11-05

**Soundness:** 2
**Presentation:** 3
**Contribution:** 2
**Rating:** 5
**Confidence:** 4

**Summary:**

This paper incorporates reward scaling with layer normalization and infeasible action penalization into existing offline reinforcement learning algorithms, specifically TBC+BC with critic ensemble, to enhance their performance. The proposed method is evaluated on various offline and offline-to-online benchmarks and compared with several baseline methods. The results show that the proposed method outperforms the baseline methods on the chosen tasks.

**Strengths:**

- The presentation is clear and easy to follow.
- The proposed method is incremental but appears effective.
- The experimental evaluation is thorough and covers a wide range of tasks.

**Weaknesses:**

- The motivation for penalizing infeasible actions is not clear. As shown in Figure 2(c) and (d), it seems penalizing OOD actions within the feasible region will lead to the same optimal policy (e.g., argmax Q) as RS-LN and PA. In this case, the proposed method is equivalent to RS-LN and PA, though the absolute Q values are different.

- Section 3 criticizes that using a critic ensemble significantly increases the training complexity. However, the proposed method also uses a critic ensemble (4 for AntMaze, 10 for MuJoCo and Adroit), which undermines the advantage of the proposed method and makes its motivation less convincing. A comprehensive comparison of the proposed method without a critic ensemble against other baseline methods would be helpful to clarify the contribution of the proposed method.

- Some of the results are not consistent with the claims in the paper. Please see the questions section for details.

**Questions:**

- Figure 3 shows results from a single seed on the toy example. Is this conclusion consistent when using different seeds? Will the shape worsen or converge to a pattern when the reward scaling factor is larger?
- Line 231 states, 'This regulated reduction in Srank prevents overfitting by minimizing the learning of irrelevant noise'. How can this conclusion be drawn from the results?
- Figure 7 shows that a larger reward scaling factor leads to better performance. Is this a general conclusion for other tasks? Is there any limitations about the reward scaling factor?
- Line 270 states, 'the performance does not heavily depend on the values of U when these values are set as 100 to 1000 times the boundary of the feasible region'. However, the results in Figure 7 show that performance decreases when U is large on 3 out of 4 tasks. Can you clarify this inconsistency?
- See Weaknesses.

I am willing to increase my score if all my concerns are addressed.

---

> ### Author Response · Authors · 2024-11-21
> **Authors' response to reviewer 6JBM**
>
> We appreciate the comments aimed at clarifying the contributions and unclear explanations of PARS, as well as suggesting ways to present more robust experimental results. Incorporating this feedback is expected to further improve the quality of the paper.
>
> &nbsp;
>
> ### **[W1] Regarding the motivation of PARS**
>
> Thank you for giving us the opportunity to further strengthen our motivation by comparing it with prior works. In our common response [C1], we have clarified our motivation and contributions more explicitly. As pointed out by the reviewer, in Figure 2, while both Penalizing OOD actions within the feasible region (c) and RS-LN and PA (d) aim to achieve the same goal of finding the max Q policy offline, **(d) demonstrates a smoother decrease in the Q-function in OOD regions. This characteristic is particularly advantageous during online fine-tuning. As we show through various figures and additional experiments in Appendix B of our revised paper, if the OOD Q-values are learned too conservatively or are fitted to small values offline, it could lead to adverse effects during online fine-tuning. A more detailed response can be found in Appendix B of the revised paper.**
>
> &nbsp;
>
> ### **[W2] Regarding the ensemble**
>
> We regret any confusion that may have arisen from our prior wording, as we did not intend to criticize the use of critic ensembles per se. Recent approaches such as MSG [Ref.3] with 64 critics and RLPD [Ref.4] with 10 critics illustrate that critic ensembles are both straightforward and effective. **Our point, however, is that relying solely on critic ensembles for regularization, as seen in methods like SAC-N [Ref.5], may demand as many as 500 critics for certain datasets, leading to an excessive complexity overhead. Moreover, even with this added complexity, SAC-N is unable to solve the AntMaze task (as also mentioned in CORL [Ref.6]).**
>
> As shown in Table R3, **for AntMaze, PARS performs well without critic ensembles**, although it shows a slight decrease in score compared to PARS with four critics.
>
> **[Table R3]** PARS AntMaze performance without critic ensembles.
>
> |  | PARS | PARS (no ensemble) |
> | --- | --- | --- |
> | umaze | 93.8 | 93.5±5.6 |
> | umaze-diverse | 89.9 | 91.7±4.0 |
> | medium-play | 91.2 | 88.8±2.5 |
> | medium-diverse | 92.0 | 87.5±1.7 |
> | large-play | 84.8 | 74.6±10.2 |
> | large-diverse | 83.2 | 81.0±9.5 |
> | ultra-play | 66.4 | 56.8±3.2 |
> | ultra-diverse | 51.4 | 46.0±19.8 |
>
> In tasks like MuJoCo and Adroit, ensemble integration proves beneficial, but a simple method significantly reduces the number of ensembles required compared to using only the ensemble approach. We envision that future research will focus on advancing network expressivity and generalization ability, leading to powerful RL algorithms that operate with minimal complexity.
>
> &nbsp;
>
> ### **[Q1] Didactic examples with more seeds and increased reward scale**
>
> We provide results for additional seeds and for setting $c_\text{reward}$ to values greater than 100, such as 10,000, in Appendix J of the revised paper. First, varying the seeds showed slight differences, but the overall trend remained consistent. Additionally, when $c_\text{reward}$ was increased, applying only LN caused the shape to change slightly depending on $c_\text{reward}$, but no clear pattern emerged. However, when both LN and PA were applied, similar patterns persisted even as $c_\text{reward}$ increased further.
>
> &nbsp;
>
> ### **[Q2] Regarding the interpretation of Srank**
>
> The effective rank of the representation, Srank, was proposed in [ref.7], which stated that a reduction in effective rank results in implicit under-parameterization and can lead to performance degradation. However, in “Small Batch Deep Reinforcement Learning (NeurIPS 2023)” [ref.8], based on previous findings that deep RL networks tend to overfit during training, it was observed that the network adapts better to an earlier rank collapse than to a later one. Specifically, when RL is trained with smaller batch sizes, the Srank collapse occurs earlier in training and remains lower compared to larger batch sizes. During this earlier collapse, performance improves more effectively.
>
> We referred to the analysis results of these two studies. As shown in Figure 4, when LN is applied, an increase in the reward scale initially causes the effective rank to decrease, but it then converges. In contrast, when LN is not applied, the effective rank continues to decrease during training as the reward scale increases. Consistent with the analysis results on small batch sizes, applying LN with an increasing reward scale also leads to more effective performance improvements. This suggests that the reduction in effective rank in this context actually prevents overfitting and promotes more effective performance enhancement.

---

> > ### Author Response · Authors · 2024-11-21
> > **Authors' response to reviewer 6JBM**
> >
> > ### **[Q3] Regarding $c_\text{reward}$**
> >
> > In Appendix H, Figure 18 shows the results of increasing $c_\text{reward}$ in the MuJoCo and Adroit settings. Unlike AntMaze, MuJoCo and Adroit feature dense rewards, so even a 10x increase in $c_\text{reward}$ yielded noticeable effects. However, in contrast to AntMaze, increasing $c_\text{reward}$ beyond 10x provided no additional benefits.
> >
> > To examine the limitations of $c_\text{reward}$, we increased $c_\text{reward}$ up to $10^{10}$ and conducted additional experiments on the antmaze-umaze-diverse dataset using five random seeds. We found that stable results can be achieved with a mean performance of 97.3, even when $c_\text{reward}$ increases up to $10^{6}$. However, performance began to decrease at $c_\text{reward} = 10^{8}$, with a mean performance of 81.3 at $c_\text{reward} = 10^{8}$ and 45.4 at $c_\text{reward} = 10^{10}$. Specifically, when $c_\text{reward}$ was $10^{8}$, the loss function increased to $10^{16}$, and when $c_\text{reward}$ reached $10^{10}$, the loss further escalated to $10^{20}$. Such a large loss scale could potentially lead to overflow or gradient explosion, highlighting the importance of carefully managing the loss scale.
> >
> > &nbsp;
> >
> > ### **[Q4] Regarding U**
> >
> > What we intended to convey regarding U is that when U is around 100–1000, it works well practically. Figure 9 (I think the Figure 7 you mentioned is a typo?) shows that when U is 100 or 1000, there is a slight difference, but it performs reasonably well. However, as the reviewer noted, performance decreases when U is 10,000, as the effect of PA diminishes when U is as far off as 10,000. The performance drop in 3 out of 4 tasks, which the reviewer mentioned, occurs specifically when U is set to this larger value of 10,000, so the claims and experimental results in our paper appear to be consistent. If this isn't the point the reviewer was curious about, feel free to let us know again.
> >
> > &nbsp;
> >
> > [Ref.3] Ghasemipour, Kamyar, Shixiang Shane Gu, and Ofir Nachum. "Why so pessimistic? estimating uncertainties for offline rl through ensembles, and why their independence matters." *NeurIPS 2022*.
> >
> > [Ref.4] Ball, Philip J., et al. "Efficient online reinforcement learning with offline data." *ICML 2023*.
> >
> > [Ref.5] An, Gaon, et al. "Uncertainty-based offline reinforcement learning with diversified q-ensemble." *NeurIPS 2021.*
> >
> > [Ref.6] Tarasov, Denis, et al. "CORL: Research-oriented deep offline reinforcement learning library." *NeurIPS 2023*.
> >
> > [Ref.7] Kumar, Aviral, et al. "Implicit Under-Parameterization Inhibits Data-Efficient Deep Reinforcement Learning." *ICLR 2021*.
> >
> > [Ref.8] Obando Ceron, Johan, Marc Bellemare, and Pablo Samuel Castro. "Small batch deep reinforcement learning." *NeurIPS 2023*.

---

> > > ### Comment · Reviewer_6JBM · 2024-11-25
> > >
> > > Thank you for your detailed responses. I appreciate the effort you put into addressing my concerns and clarifying your work. While your responses have resolved some of my questions, I still believe that the paper is not ready for publication in its current form.
> > >
> > > After careful consideration, I have decided to maintain my original score.

---

> > > > ### Author Response · Authors · 2024-11-25
> > > >
> > > > We appreciate your thoughtful review of our work and responses. If there are any unresolved issues in our responses, we would be happy to discuss them further and make improvements to the paper.
> > > >
> > > > Additionally, through discussions with other reviewers, we confirmed that the idea of PARS can also be applied effectively to in-sample learning-based methods (Appendices F.2 and D.2). Furthermore, we analyzed the reasons behind its particular effectiveness when combined with TD3+BC (Appendix K). These points have been incorporated into the revised paper.
> > > >
> > > > If there are additional discussion points or areas where the paper can be further improved, please feel free to let us know!

---

### Official Review · Reviewer_ypm4 · 2024-11-09

**Soundness:** 3
**Presentation:** 3
**Contribution:** 2
**Rating:** 6
**Confidence:** 4

**Summary:**

The paper introduces the PARS algorithm, which aims to address Q-value extrapolation errors prevalent in offline reinforcement learning (RL). The authors propose a twofold approach combining (1) Reward Scaling with Layer Normalization (RS-LN) and (2) Penalizing Infeasible Actions (PA). The former stabilizes the Q-function learning by scaling rewards while employing layer normalization to manage output variances effectively. The latter penalizes Q-values for infeasible action regions to prevent overestimation outside the feasible data coverage. The paper shows that PARS, built upon the minimalist TD3+BC algorithm, outperforms existing state-of-the-art (SOTA) methods in offline, and offline-to-online setups, particularly in the challenging AntMaze environment.

**Strengths:**

- The paper demonstrates the effectiveness of the proposed PARS method through comprehensive empirical results across various domains, showing that it outperforms or matches strong baselines.
- The method’s simplicity and minimal code changes required for implementation make it attractive for practitioners looking to enhance existing RL models without significant overhead.
- The paper is well-written, with clear visualizations that effectively convey the concepts and intuitions behind the approach. This aids in understanding the methodology and its application.

**Weaknesses:**

- The way infeasible regions and reward scaling are defined—through arbitrary thresholds without adaptive or data-driven characterization—limits the method's novelty.
- While the paper mentions following prior work for hyperparameter tuning, the strategy of tuning hyperparameters with online interactions contadicts offline RL, which undermines the problem’s nature. Using Off-Policy Evaluation methods for hyperparameter selection, or fixing a set of hyperparameters for each environment would be better aligned with Offline RL.
- The paper does not discuss offline RL methods that do not query out-of-distribution (OOD) actions, such as SQL [1] , XQL [2],  and IQL. and how PARS ideas can be leveraged by those.

[1] Offline RL with no OOD actions: In-sample learning via implicit value regularization. ICLR 2023.

[2]  Extreme q-learning: Maxent rl without entropy. 2023

**Questions:**

- Please check weaknesses.
- How do you define the feasible boundary for the action space in your method? Specifically, what do  l_i  and  u_i  represent? Are they derived as the minimum and maximum a_i within the dataset?

---

> ### Author Response · Authors · 2024-11-21
> **Authors' response to reviewer ypm4**
>
> We appreciate the reviewer for their positive comments on the strengths of PARS, such as its simplicity and effectiveness, as well as for highlighting further promising discussion points about PARS.
>
> &nbsp;
>
> ### **[W1] Regarding hyperparameters**
>
> Thank you for pointing out an important point. In RL, unlike in NLP or vision tasks, there is no clear validation set or environment. Therefore, offline evaluation metrics that can be utilized without online evaluation would be highly advantageous. However, as the reviewer mentioned, many prior works ([Ref.1,2]) allowed a certain degree of hyperparameter tuning, and we followed these works for tuning. Nevertheless, we agree with the reviewer’s point and **further verified that comparable performance in the AntMaze domain can be achieved with just a single hyperparameter setting.** This is particularly significant because many offline algorithms have struggled in the AntMaze domain, especially in the Ultra environment. Achieving significantly better performance than the baseline with a single hyperparameter setting in the AntMaze domain—even though it is slightly lower than the PARS score presented in the paper—underscores a key advantage of PARS. The hyperparameters we used for the single-hyperparameter experiment are: reward scale of 10,000, alpha of 0.001, and beta of 0.01.
>
> **[Table R1]** PARS AntMaze performance using a single hyperparameter setting.
>
> |  | other SOTA | other SOTA (goal-conditined) | PARS | PARS (single hyperparam) |
> | --- | --- | --- | --- | --- |
> | umaze | 97.9 (MSG) | 91.6 (GC-IQL) | 93.8 | 93.3±3.2 |
> | umaze-diverse | 88.3 (ReBRAC) | 88.8 (GC-IQL) | 89.9 | 84.7±5.7 |
> | medium-play | 85.9 (MSG) | 84.1 (HIQL) | 91.2 | 89.4±2.9 |
> | medium-diverse | 84.6 (MSG) | 86.8 (HIQL) | 92.0 | 86.2±5.6 |
> | large-play | 64.3 (MSG) | 86.1 (HIQL) | 84.8 | 77.5±1.8 |
> | large-diverse | 71.2 (MSG) | 88.2 (HIQL) | 83.2 | 83.5±3.6 |
> | ultra-play | 22.4 (ReBRAC) | 56.6 (GCPC) | 66.4 | 60.2±7.6 |
> | ultra-diverse | 14.2 (IQL) | 54.6 (GCPC) | 51.4 | 50.8±11.1 |
>
> &nbsp;
>
> ### **[W2] Comparison with in-sample learning-based methods**
>
> Thank you for suggesting an insightful discussion point. Based on your suggestion, we analyzed how the ideas of PARS could be applied to a representative in-sample learning-based method, IQL, and **included the analysis in Appendix F.2 of the revised paper.** As shown in Appendix F.2, since IQL does not perform OOD sampling, PA does not provide benefits for this method. Conversely, RS-LN demonstrated noticeable effectiveness, highlighting its potential for broader applicability and impact when integrated into in-sample learning-based approaches.
>
> &nbsp;
>
> ### **[Q1] Regarding feasible bound**
>
> In this work, all the domains we benchmarked have a feasible action range of -1 to 1. Consequently, we conducted experiments with $L_i = -1$ and $U_i = 1$. We believe this does not introduce additional assumptions compared to other algorithms, as they also leverage the knowledge of the feasible action range being -1 to 1 and typically use a tanh activation function in the actor network as a standard practice.
>
> However, in practical scenarios, as pointed by the reviewer, assuming an unknown feasible action range could further improve the algorithm's scalability. To explore this, as suggested by the reviewer, we conducted additional experiments by extracting the minimum and maximum bounds for each action dimension from the dataset. As shown in Table R2, training with the feasible bounds calculated from the dataset resulted in no difference in performance. This is likely because we impose the infeasible penalty not near the feasible bounds but at a point significantly farther away, making it less critical to know the exact values and feasible to use the calculated ones.
>
> **[Table R2]** PARS with a calculated feasible bound.
>
> |  | PARS | PARS (calculated feasible bound) |
> | --- | --- | --- |
> | umaze | 93.8 | 96.4±1.5 |
> | umaze-diverse | 89.9 | 88.1±6.9 |
> | medium-play | 91.2 | 89.8±5.1 |
> | medium-diverse | 92.0 | 91.7±3.8 |
> | large-play | 84.8 | 82.4±7.1 |
> | large-diverse | 83.2 | 84.6±5.9 |
> | ultra-play | 66.4 | 62.2±14.2 |
> | ultra-diverse | 51.4 | 53.3±9.9 |
>
> &nbsp;
>
> [Ref.1] Tarasov, Denis, et al. "Revisiting the minimalist approach to offline reinforcement learning." *NeurIPS 2023*.
>
> [Ref.2] Nikulin, Alexander, et al. "Anti-exploration by random network distillation." *ICML* 2023.

---

> ### Author Response · Authors · 2024-11-26
>
> Thank you once again to the reviewer for your thoughtful comments on our work. In our new common response, we have summarized the paper revisions made during the rebuttal period. We would be delighted if our revised paper provides a clearer understanding of our work.
>
> In particular, we have added content on [W2] comparison with in-sample learning-based methods in Appendix F.2, explained why RS-LN can also be effective in in-sample learning-based methods in Appendix D, and discussed its particular effectiveness in TD3+BC in Appendix K. Other responses can be found in our previous responses. Please feel free to reach out if you have any additional questions.

---

### Author Response · Authors · 2024-11-21
**Common response**

We sincerely thank all the reviewers for their detailed comments on our work. The reviewers' feedback has provided us with a valuable opportunity to address unclear points and strengthen the contributions of our work. In this common response, we address questions that reviewers have commonly raised. In particular, we have clarified our motivation and contributions and highlighted the revisions made in the updated paper.

&nbsp;

### **[C1] Motivation and contributions of PARS**

Our goal with PARS is to *"design an RL algorithm that performs well both offline and during the online fine-tuning phase, while utilizing reduced complexity in methods and implementation.”*

For strong offline performance, some previous works have aimed to lower Q-values for OOD actions by effectively distinguishing between ID and OOD regions and employing various methods to reduce Q-values within the OOD region. **In contrast, PARS does not directly enforce lower Q-values for OOD actions within the feasible region, due to the challenges of precisely reducing values in the OOD region and the potential side effects during the online fine-tuning phase. Instead, it aims to fully leverage the network's expressivity and adaptability, which naturally results in a reduction in Q-values.**

- Other methods often establish a clear boundary between ID and OOD regions, whereas **PARS maintains smoothness at the ID/OOD boundary**, enabling flexible exploration and better adaptation to distribution shifts during online fine-tuning phase.
- PARS is very simple yet highly effective across domains, dataset qualities, and training phases.

In the manuscript's Section 3, we compared prior works that aimed to perform critic regularization with PARS. Additionally, since our goal is not just to design an algorithm that performs well during the offline phase but also excels in the online fine-tuning phase, we provide a detailed explanation in the revised paper's Appendix B on how PARS offers advantages over other prior algorithms from the perspective of online fine-tuning.

&nbsp;

### **[C2] Paper modification**

The sections that were revised or added in the revised paper are as follows. The revised parts are highlighted in blue. Additionally, typos and unclear words pointed out by the reviewers have also been corrected.

- [modified] Section 3. Critic Regularization For OOD Actions
    - We clarify the differences between methods that rely solely on the critic ensemble and those that perform OOD penalization within the feasible region, ensuring that the advantages of RS-LN and PA are more distinctly highlighted.
- [Added] Appendix B.  In-Depth Comparison with Prior Works
    - Compared to prior works, we analyze the strengths of PARS in the context of online fine-tuning, supported by figures and additional experiments.
- [Added] Appendix D.  In-Depth Analysis of RS-LN Effectiveness
    - We provide an analysis of the potential reasons behind the effectiveness of RS-LN.
- [Added] Appendix F.2. Extension to In-Sample Learning-Based Methods
    - We present results demonstrating the extension of the PARS concept to in-sample learning-based methods.
- [Added] Appendix J. Didactic Example with More Seeds and Increased $c_\text{reward}$

----- Additional Updates -----

- [Added] Appendix B.2. More Related Works
- [Added] Appendix D.2. Comparison of TD3+BC and IQL Critic Updates and the Effectiveness of RS-LN
- [Added] Appendix K. Why is the combination of TD3+BC with RS-LN and PA effective?

---

### Author Response · Authors · 2024-11-26
**A summary of further paper revisions during the rebuttal period**

Dear Reviewers,

Thanks to the reviewers' ongoing feedback and dedication to our work, various improvements have been made to the paper during the rebuttal period. We would like to summarize these updates. Through extensive discussions, **we believe the paper has been updated to better convey the motivation behind PARS, the reasons for its strong effectiveness, and its versatility.** The potential to develop a simple yet powerful RL algorithm validated through PARS is expected to make RL more practical and inspire a variety of future works.

As the rebuttal period has been extended, we are willing to address and incorporate any remaining discussion points.

- **Regarding Motivation → Section 3 & Appendix B**

    We discuss the advantages of RS-LN and PA over other prior methods during both the offline phase and online fine-tuning. Appendix B specifically delves into the effectiveness of these methods in the context of online fine-tuning.

- **Regarding the Versatility of the PARS Idea → Appendix F**

    We apply the concept of PARS to in-sample learning-based methods and confirm that RS-LN is also effective for such methods.

- **Regarding the Effectiveness of RS-LN → Appendix D**

    We explore the potential reasons behind the effectiveness of RS-LN and discuss how RS-LN can be universally effective for TD3+BC and in-sample learning-based algorithms such as IQL.

- **Regarding the Combination of TD3+BC with RS-LN & PA → Appendix K**

    We discuss why RS-LN and PA show greater effectiveness specifically with TD3+BC compared to in-sample learning-based methods like IQL.


We look forward to further discussions and welcome any additional feedback to enhance the paper.

---

### Meta-Review · Area_Chair_hHcV · 2024-12-23

**Metareview:**

This paper argues that reward scaling (multiplying the reward by a scalar) with layer normalization in the value network, in addition to penalizing the value of infeasible actions can be an effective technique for offline reinforcement learning. The paper discusses experimental results on a number of standard benchmark problems.

The discourse with the reviewers revolved around the details of this simplistic approach, e.g., why is it necessary to penalize infeasible actions if they are never going to be sampled, and whether reward scaling can really be this effective. In this discussion, a few points seem to have been missed.

First, multi-layer perceptrons are biased, at early stages of training, to low-frequency functions. This is why penalizing the value for infeasible actions makes the predicted value function smoother (it fills in the gaps at intermediate actions regardless of whether they are good or bad). As such, smoother values need not necessarily be good for exploration or offline RL. If one trains for more iterations, this bias is rectified, and therefore one will not see smooth value functions. Altogether, it is not clear why one would impose this penalty. Second, the whitening occurring in layer normalization helps in fitting MLPs more quickly, it is not clear whether from the evidence in this paper whether it has anything to do with offline RL. Reward scaling is an interesting idea, but without further evidence of _why_ this particular approach helps in offline RL (perhaps it does so because scaling up the loss helps the user control the relative magnitude of extrapolation bias), it is difficult to argue for this paper. I would encourage the authors to investigate why these ideas work in the specific context of offline RL. The didactic examples and toy examples in the Appendix are very interesting, but not directly related to the problems being investigated in this paper.

**Additional Comments On Reviewer Discussion:**

Reviewer ypm4 was concerned about doing model selection/early stopping using online episodes and the number of new hyper-parameters introduced in this approach (action bounds, thresholds of what counts as OOD and what does not). The authors conducted some new experiments to check that with fixed hyper-parameter values for reward scaling and layer norm, they can get reasonable results on a number of antmaze problems.

Reviewer 6JBM argued that since actions never go out of the feasible set, it does not make sense to penalize infeasible actions (I agree with this point...and this is one of the two contributions of this paper). The authors conducted a toy experiment on a one-dimensional Q learning problem to argue that the estimated value function is smoother within the feasible region if one penalizes the infeasible actions. I do not see what such a smooth value function has to do with improved exploration in general. I also checked these experiments, and would argue that this is really because of the low-frequency bias of multi-layer perceptrons. In other words, if one were to train the MLP for more iterations, this low-frequency bias would vanish.

Reviewer 6JBM was also concerned about reward scaling. Effectively, the approach involves scaling up the reward to quite high values (presumably, to keep it higher than the erroneous value of missing actions) and then using layer norm in the value network to control the magnitude of the predictions. The authors did experiments with reward scaling as high as 10^10, which is very surprising.

Reviewer UZZ9 wanted to know plausible explanations for why the approach works, along with some more detailed questions about reward scaling, sparse vs dense rewards etc. The authors have provided an elaborate response to these comments, in line with their response to similar questions raised by other reviewers.

Reviewer h6w1 pointed out similarities with some existing work, and wanted the authors to clarify how their approach is different from these works. They also wanted to know how general the idea of reward scaling followed by layer normalization can be. The authors had an elaborate discussion with the Reviewer.

---

### Decision · Program_Chairs · 2025-01-22

Reject